# Privacy Budget Tailoring in Private Data Analysis

**Daniel Alabi**                                                           *alabid@cs.columbia.edu*
*Columbia University*

**Chris Wiggins**                                                   *chris.wiggins@columbia.edu*
*Columbia University*

**Reviewed on OpenReview:** *https://openreview.net/forum?id=SnPEhMyuYX*

## Abstract

We consider the problem of learning differentially private linear and logistic regression models that do not exhibit disparate performance for minority groups in the data. Small-sized datasets pose a challenging regime for differential privacy; that is, satisfying differential privacy while learning models from data can lead to models with worse accuracy for minority—in size—subgroups. To address this challenge, inspired by Abowd & Schmutte (2018), we propose: (i) to *systematically* tailor the privacy budget to the different groups, (ii) use linear optimization oracles in a grid to optimize Lagrangian objectives that correspond to fair learning and optimization. We present efficient differentially private algorithms for linear and logistic regression subject to fairness constraints (e.g., bounded group loss) that allocate the privacy budget based on the private standard error of each subgroup in the data. Consequently, the formulation reduces the amount of noise added to these groups, which leads to more accurate models for such groups. We validate the proposed, group-aware budget allocation, method on synthetic and real-world datasets where we show significant reductions in prediction error for the smallest groups, while still preserving sufficient privacy to protect the minority group from re-identification attacks. In addition, we provide sample complexity lower bounds for our problem formulation.

## 1 Introduction

Learning predictive models that satisfy differential privacy (DP) (Dwork et al., 2006)—a worst-case cryptographic privacy definition—is an increasingly important requirement in healthcare (Pfohl et al., 2019), social science (Evans et al., 2023), and federated learning (McMahan et al., 2017). Overparametrized deep models increasingly memorize portions of the input training data (Carlini et al., 2020); however, such memorization behavior is neutralized when these models are optimized via differentially private procedures (Carlini et al., 2018; Chen et al., 2020). Despite such benefit, Bagdasaryan et al. (2019) show that models optimized to satisfy DP via privatized versions of (stochastic) gradient descent (DPSGD) (Bassily et al., 2014; Chaudhuri et al., 2011; Abadi et al., 2016) have worse performance—test accuracy for example—for minority (in size) groups in the data if the privacy budget is split equally amongst all groups. Disparate performance due to DP hampers its deployment in high-stakes applications like healthcare (Santos-Lozada et al., 2020) and synthetic census data release (Petti & Flaxman, 2019; Kenny et al., 2021). Consequently, *we consider the challenge of learning differentially private models—from data via (stochastic) gradient descent—that do not exhibit disparate performance for minority subgroups in the data.*

The key insight in our proposed approach is to *apportion the privacy budget across different subgroups in the data.* We propose a two-stage group-aware scheme in which we instantiate this insight in the linear regression setting. The proposed algorithm can handle the case where an individual can be in multiple groups. While we instantiate our formulation for linear regression, the key idea of allocating the privacy budget across sub-groups in the data can be applied more broadly.

First, we estimate the standard error (SE) of the prediction across sub-groups in the data for a 'precursor' model. We then assign the privacy budget for DP according to the estimated SE of the predictions—allocating more privacy budget to subgroups with higher standard error. Larger standard errors in a particular group suggests a form of underfitting and structural uncertainty in the group statistic; hence, larger DP noise addition to such groups exacerbates the uncertainty in the parameter estimates for these groups.

We justify the group-aware privacy budget allocation with a theoretical analysis of an idealized 2-mixture of Gaussians model (in Lemma 3.1). We analyze the expected loss of a randomized linear regression coefficient vector on the mixture model, and show that naively applying differential privacy disadvantages the group with higher standard prediction error—typically the smaller group.

In the second stage, we use the privacy budget allocation to learn DP linear regression estimates via gradient descent. Private versions of gradient descent typically involve gradient clipping and calibrated Gaussian noise addition. Group-aware privacy budget allocation based on standard errors of the prediction across groups results in less noise addition to groups with larger standard errors of the prediction.

A potential objection to the proposed formulation is that relaxing the privacy requirement for minority subgroups will make these input groups more susceptible to privacy attacks. Indeed, this could be the case if caution is not exercised. However, recent empirical findings show that models produced by private (stochastic) gradient descent with even minimal levels of privacy (budget of $\epsilon=89$) are still effective at stemming re-identification attacks (Carlini et al., 2018). Also, as suggested by one of the reviewers of this work, it might be better to set a minimum or target level for the privacy budget for the minority subgroup(s) and then calculate an overall budget for the end-to-end scheme. We leave this choice to the framework designer. Also, since we build upon previous work on Lagrangian fair regression (Agarwal et al., 2019), the majority group might incur a slightly larger error so that the minority groups could gain in predictive accuracy. As per DP guarantees, we operate with zero-Concentrated differential privacy (zCDP) and track the budget of our proposed mechanisms with $\rho$. A mechanism that satisfies $\rho$-zCDP also satisfies $(\epsilon, \delta) = (\rho + 2\sqrt{\rho \log(1/\delta)}, \delta)$-DP for all $\delta > 0$, and equivalently, $\epsilon$-DP satisfies $\epsilon^2/2$-zCDP (Bun & Steinke, 2016).

The two-stage formulation that we present can also be extended to incorporate user-defined 'fairness' objectives like bounded sub-group loss. To accomplish this, we consider a linear regression formulation subject to bounded subgroup loss (BGL), and show how to incorporate the constraint into a reformed objective using the Lagrangian approach of Agarwal et al. (2019). We then solve the Lagrangian formulation using the group aware approach we proposed here.

We evaluate the performance of the two-stage group-aware DP linear regression formulation and the fair constrained versions across synthetic and real world datasets. We find that the proposed two-stage formulation results in a 10% to 50% reduction in the mean standard prediction error for the smallest groups in the data both for traditional differentially private learning and the fair Lagrangian setting. Taken together, the empirical results suggest that group-aware allocation of the privacy budget can ease the disparate utility impact of differential privacy.

A series of closely related works (Xu et al. (2020), Abowd & Schmutte (2018), and Tran et al. (2021a)) consider a setting similar to ours. Xu et al. (2020) propose DPSGD-F, which adaptively performs gradient clipping and noise addition to groups in the data. Abowd & Schmutte (2018) allocate the privacy parameter to subgroups in the data, but focus on estimating the resulting economic costs of such decision. In particular, Abowd & Schmutte (2018) treat privacy budget allocation as a resource allocation problem via an economic lens: choose budgets such that the marginal cost of increasing privacy equals the marginal benefit. In our case, we just use statistical properties of the dataset itself for resource allocation. Another related work is that of Tran et al. (2021a) which studies the underlying cause of disparate impact for output perturbation and DP-SGD, and find that the noise due to privacy, gradient clipping, and a group difficulty term captured by the group-based trace of the hessian of the loss govern the underlying group disparities. These findings align with ours, but we focus on the privacy budget, which is the fundamental parameter that determines the level of noise added to each group.

**To conclude, we summarize our contributions as follows:**

- We present a two-stage, end-to-end differentially private, scheme for linear regression that maintains 'reasonable' performance for minority subgroups in the data. To do this, we allocate the privacy budget towards different subgroups in the data.

- We conduct extensive empirical evaluation of the two-stage formulation for synthetic and real world data sets and observe 10% to 50% reduction in mean square standard prediction error for the smallest groups across settings.

## 2 Background and Linear Regression with DP

In this section, we provide an overview of notation, definitions, and problem formulation.

**Setup.** Let $\mathcal{X}$ denote the domain of non-sensitive features, $\mathcal{A}$ the domain of sensitive attributes, and $\mathcal{Y}$ correspond to the set of possible labels. We take $\mathcal{C}_P$ to denote a decision set from which a regression coefficient vector $\theta$ is chosen. For example, $\mathcal{C}_P$ can be the space of $P$-dimensional real vectors i.e., $\mathcal{C}_P = \mathbb{R}^P$. We use the random variable $(X, A, Y)$ to specify a joint distribution over $(\mathcal{X}, \mathcal{A}, \mathcal{Y})$ and $X$, $A$, $Y$ to represent marginal distributions over $\mathcal{X}, \mathcal{A}, \mathcal{Y}$ respectively. Given any dataset $Z$ with row size $n$, we can categorize by sensitive and insensitive attributes as follows: $Z = (Z_{\mathcal{X}}, Z_{\mathcal{Y}}, Z_{\mathcal{A}}) = (Z_{\mathcal{X}\mathcal{Y}}, Z_{\mathcal{A}})$ where $Z_{\mathcal{A}} \in \mathcal{A}^n, Z_{\mathcal{X}\mathcal{Y}} \in (\mathcal{X} \times \mathcal{Y})^n$. For example, $\mathcal{A}$ could be the set of all possible outcomes on a HIV medical test: $\{+, -\}$. Obviously, due to state and federal privacy law, the results of a HIV test cannot be released without permission or without some form of explicit privacy protection mechanisms; thus, $\mathcal{A}$ is sensitive. On the other hand, zip code might be considered a non-sensitive attribute.

**Remark.** We assume that the dataset size, $n$, (and the size of all subgroups) is public information.

The notion of privacy that we consider in this work is record-level (bounded) differential privacy. Our framework can also be instantiated in the unbounded (add-remove instead of swap) DP model without much modification.

**Definition 2.1** (Zero-Concentrated differential Privacy (Bun & Steinke, 2016))**.** *An algorithm $\mathcal{M} : \mathcal{Z}^n \to \mathcal{R}$ is $\rho$ **zero-concentrated differentially private** (zCDP) if for all neighboring $z', z \in \mathcal{Z}^n$, $D_{\gamma}(\mathcal{M}(z), \mathcal{M}(z')) \leq \rho\gamma$, for all $\gamma \in (1, \infty)$, where $D_{\gamma}(A\|B)$ is the $\gamma$-Rényi divergence between random variables $A$ and $B$. The randomness is over the coin flips of the algorithm $\mathcal{M}$. The privacy budget allocated to all groups is $\rho$.*

Definition 2.1 reduces to standard pure DP (Dwork et al., 2006) as $\gamma \to \infty$.

**Fairness Definitions.** For this paper, we will focus on a single notion of fairness: *Bounded Group Loss* (BGL). Intuitively, BGL requires that the prediction error for any protected group remain below a certain pre-specified level. A more formal definition is given in Definition 2.2.

**Definition 2.2** (Bounded Group Loss (Agarwal et al., 2019))**.** *Let $\theta$ be a predictor over $(X, A, Y)$. Then $\theta \in \mathcal{C}_P$ satisfies **Bounded Group Loss** if $\forall a \in \mathcal{A}$, $\mathbb{E}[\ell(Y, \theta(X)) \mid A = a] \leq \eta_a$, where $\ell : \mathcal{Y} \times \mathcal{X} \to \mathbb{R}$ is a loss function we wish to minimize and $\eta_a$ is a loss constraint for group $a \in \mathcal{A}$.*

**Linear Regression with Differential privacy (e.g., see (Alabi & Vadhan, 2022; Alabi, 2022) and references therein).** To set the stage, we work through DP Gradient Descent for linear regression. Given a dataset $Z = \{(\mathbf{x}_i, y_i)\}_{i=1}^n$, suppose that $\mathcal{C}_P \subseteq \mathbb{R}^P$. We consider a function, $h : \mathcal{C}_P \to \mathbb{R}$, defined as $h(\theta) = \frac{1}{n} \sum_{i=1}^n (\langle \theta, \mathbf{x}_i \rangle - y_i)^2$ with gradient: $\nabla h(\theta) = \frac{2}{n} \sum_{i=1}^n \mathbf{x}_i (\langle \theta, \mathbf{x}_i \rangle - y_i) \in \mathbb{R}^P$. To optimize $h$ with zCDP, we add Gaussian noise to the gradients $\nabla h$ while optimizing for $\theta$. This formulation requires two important hyper-parameters: (1) $T$: the number of gradient calls to $\nabla h$, and (2) $\Delta$: clipping bound. The number of iterations determines how the privacy budget $\rho$ is apportioned. For any neighboring databases $z, z'$ and query $q$, if $|q(z) - q(z')| \leq \Delta$ for a finite $\Delta$, then we can satisfy $\rho$-zCDP. In general, gradients can be unbounded so to satisfy zCDP, we clip the gradients to lie in a certain range.

## 3 Group-Based Privacy Budget Allocation & Langrangian Fair Regression

In this section, we provide justification for the need to allocate the privacy budget in DP to groups in the data in order to mitigate disparate model performance. To do this, we consider a mixture of 2 Gaussians and

analytically derive the expected prediction risk for the case where a single predictor is learned from data. This analysis provide insight into how DP regression exacerbates performance disparity. Consequently, we arrive at the natural insight that tailoring the privacy budget to groups in the data based on the standard error of an initial—precursor—predictor helps mitigate performance disparity. In the second subsection, we describe the Lagrangian formulation for fair regression.

## 3.1 Mixture Model Analysis for Standard Error

Consider a mixture model of two groups, for a linear model, where the output for each group is derived as: $Y_1 \sim \mathcal{N}(X_1\beta_1, \sigma_1^2 I_{n_1 \times n_1})$, $Y_2 \sim \mathcal{N}(X_2\beta_2, \sigma_2^2 I_{n_2 \times n_2})$, and where $X_1 \in \mathbb{R}^{n_1 \times d}, X_2 \in \mathbb{R}^{n_2 \times d}$ are the design matrices. $Y_1 \in \mathbb{R}^{n_1}, Y_2 \in \mathbb{R}^{n_2}$ are the dependent variables. In this model, $\sigma_1, \sigma_2$ characterize the inherent noise in the dependent while $\beta_1, \beta_2 \in \mathbb{R}^d$ are the regression coefficients.

Suppose we wish to learn a single linear predictor, $\beta$, for the mixture model $c \cdot \mathcal{N}(X_1\beta_1, \sigma_1^2 I_{n_1 \times n_1}) + (1-c) \cdot \mathcal{N}(X_2\beta_2, \sigma_2^2 I_{n_2 \times n_2})$ for any $c \in (0,1)$. Learning predictors with low error, in this case, can be achieved via randomization. In particular, to learn a single predictor when group membership is unknown, one predictor is to predict $\beta_1$ with probability $c$ and $\beta_2$ with probability $(1-c)$. Now we state the expected square loss for this predictor.

**Lemma 3.1.** *For any $c \in (0,1)$, $\beta_1, \beta_2 \in \mathbb{R}^d$, $X_1 \in \mathbb{R}^{n_1 \times d}, X_2 \in \mathbb{R}^{n_2 \times d}$, $\sigma_1, \sigma_2 > 0, n_1, n_2 \in \mathbb{Z}_+$, define the mixture model $Y \sim c\mathcal{N}(X_1\beta_1, \sigma_1^2 I_{n_1 \times n_1}) + (1-c)\mathcal{N}(X_2\beta_2, \sigma_2^2 I_{n_2 \times n_2}) \sim cY_1 + (1-c)Y_2$. Let $\beta = \beta_1$ with probability $c$ and $\beta = \beta_2$ with probability $1-c$, then the prediction $\hat{Y}$ is $Y_1$ with probability $c$ and $Y_2$ with probability $(1-c)$. The expected prediction loss of $\beta$ on $Y$ is $c^2 \cdot n_1 C \sigma_1^2 + (1-c)^2 \cdot n_2 C \sigma_2^2 + c(1-c)\Gamma_1 + c(1-c)\Gamma_2$ where $\Gamma_1 = \|Y_2 - X_2\beta_1\|^2, \Gamma_2 = \|Y_1 - X_1\beta_2\|^2$, $C \sim \chi_1^2$, and $\chi_k^2$ is the chi-squared distribution with $k$ degrees of freedom.*

We refer to the appendix for a proof of the Lemma. As shown, the expected square loss for the optimal predictor is $c^2 \cdot n_1 C \sigma_1^2 + (1-c)^2 \cdot n_2 C \sigma_2^2 + c(1-c)\Gamma_1 + c(1-c)\Gamma_2$, where $\Gamma_1 = \|Y_2 - X_2\beta_1\|^2, \Gamma_2 = \|Y_1 - X_1\beta_2\|^2$, $C \sim \chi_1^2$, and $\chi_1^2$ is the chi-squared distribution with one degree of freedom. The errors from $c^2 \cdot n_1 \chi_1^2 \sigma_1^2$ and $(1-c)^2 \cdot n_2 \chi_1^2 \sigma_2^2$ can be controlled if $\sigma_1$ or $\sigma_2$ is small. On the other hand, the upper bounds of $\Gamma_1$ and $\Gamma_2$ depend on how much $\beta_1$ differs from $\beta_2$ based on fit to either $Y_1, Y_2$.

**Why allocate privacy budget based on standard errors?** Consider a dataset consisting of two groups $G_1 = (X_1, Y_1), G_2 = (X_2, Y_2)$ where $G_1$ has strictly larger size than $G_2$ and $G_1$ has smaller prediction standard error than $G_2$. We wish to minimize the $\ell_2$ loss of a differentially private regression estimate via DP gradient descent, across these groups, which is equivalent to minimizing: $L_1(\hat{\beta}, \Delta, \rho_1, n_1) = \frac{1}{n_1}\sum_{i=1}^{n_1}(Y_{1i} - \langle X_{1i}, \hat{\beta}\rangle)^2 + N_1(\Delta, \rho_1, n_1) = L(\hat{\beta}, X_1, Y_1) + N_1(\Delta, \rho_1, n_1)$ and $L_2(\hat{\beta}, \Delta, \rho_2, n_2) = \frac{1}{n_2}\sum_{i=1}^{n_2}(Y_{2i} - \langle X_{2i}, \hat{\beta}\rangle)^2 + N_2(\Delta, \rho_2, n_2) = L(\hat{\beta}, X_2, Y_2) + N_2(\Delta, \rho_2, n_2)$ where $N_1(\Delta, \rho_1, n_1)$ and $N_2(\Delta, \rho_2, n_2)$ are the random variables that ensure that the losses (and their gradients) satisfy zCDP — for DPGD. Here $N_1(\Delta, \rho, n_1) \sim \mathcal{N}(0, \Delta^2/(2\rho n_1))$ can be used for a $\rho$-zCDP release if $\hat{\beta}$ is public and $\Delta$ is a bound on the per-example squared residuals. Up to scaling factors, $L_1$ & $L_2$ correspond to the standard errors for $G_1$ & $G_2$ respectively.

To illustrate the source of the disparate impact incurred due to DP, let $\rho_1 = \rho_2 = \rho/2$ and suppose that the estimate $\hat{\beta}$ is computed on the entire dataset. *Without fairness constraints, the optimizer will aim to minimize average loss by prioritizing minimizing loss on the larger group or the group with smaller standard errors*, resulting in $\hat{\beta}$. Further with DP, the noise added will be agnostic to the relative sizes or standard errors across groups. As a result, for a fixed privacy budget of $\rho = \rho_1 + \rho_2$, we have the inequality, with high probability: $L_1(\hat{\beta}, \Delta, \rho_1, n_1) - L(\hat{\beta}, X_1, Y_1) \leq L_2(\hat{\beta}, \Delta, \rho_2, n_2) - L(\hat{\beta}, X_2, Y_2)$, which could result in disparate impact that is not necessarily present in the non-private case.

*To resolve the disparate impact issue, we need mechanisms that will make $N_1(\Delta, \rho_1, n_1)$, a randomized function of only two tunable quantities $(\Delta, \rho)$, strictly greater than $N_2(\Delta, \rho_2, n_2)$. As a result, we can either decrease $\Delta$ — which could result in further statistical measurement error — or increase the privacy budget $\rho_1$ or $\rho_2$. Furthermore, the "right" scaling for $\rho_1$ versus $\rho_2$ should alleviate the non-private standard errors disparity in $L_1, L_2$, thus justifying our allocation strategy by scaled standard errors. This strategy might increase the average loss for the larger group (provably relatively small increase for larger groups which goes to 0 as $n \to \infty$) but decrease the average loss for the minority group. However, as we will see in our empirical*

results we find, perhaps surprisingly, that our proposed strategy does not **always** hurt the performance in the majority group.

Using a simplified generative model, Lemma 3.1 illustrates that a mixture model has expected prediction error that scales with the standard deviation of each identifiable group. This informs our privacy allocation strategy based on standard errors. Concretely, a differentially private release of $\Gamma_2$ is $F_1(\Delta, \rho) = \sum_{i=1}^{n_1}(Y_{1i} - \langle X_{1i}, \beta_2 \rangle)^2|_0^\Delta + N_1(\Delta, \rho)$ where $|_0^\Delta$ represent a clipping operation to lie between 0 and $\Delta$ and $N_1(\Delta, \rho)$ is the noise distribution (for DP) that is a function of the clipping parameter $\Delta$ and the privacy parameter $\rho$. For example, $N_1(\Delta, \rho) \sim \mathcal{N}(0, \Delta^2/(2\rho))$ is a $\rho$-zCDP release for clipping parameter $\Delta$. A similar calculation can be done for releasing $\Gamma_1$ and quantifying the release via $F_2(\Delta, \rho) = \sum_{i=1}^{n_2}(Y_{2i} - \langle X_{2i}, \beta_1 \rangle)^2|_0^\Delta + N_2(\Delta, \rho) = A + N_2(\Delta, \rho)$. For large enough clipping parameter, the estimator $F_2(\Delta, \rho)$ will be more accurate if $N_2(\Delta, \rho)$ does not overwhelm the signal in $A$. This is more likely to happen if the standard error (and/or the sample size) in the group is large enough.

## 3.2 Solving the Lagrangian Formulation of Fair Regression

We now setup the langrangian formulation for fair linear regression, which fits under the constrained group-objective optimization formulation of Agarwal et al. (2019). In Section 4, we present algorithms to optimize this formulation.

**Setup and Notation.** Given a dataset, $Z = (Z_\mathcal{X}, Z_\mathcal{Y}, Z_\mathcal{A})$, with $K = |\mathcal{A}|$ groups, and a hypothesis class, $\mathcal{C}_P$, the Lagrangian formulation for fair regression requires: 1) **Group Losses**: The function $\ell : \mathcal{C}_P \times (\mathcal{X} \times \mathcal{Y} \times \mathcal{A})^n \to [0,1]^K$ that gives the average loss for each group for a predictor $\theta \in \mathcal{C}_P$ on the dataset $Z$; 2) **Total Group-Agnostic Loss**: $f : [0,1]^K \to \mathbb{R}$ which gives the average loss for all groups; 3) **Group-Deviation Loss**: $g : [0,1]^K \to \mathbb{R}$ a measure of loss deviation across groups.

Consider the composition of functions $f \circ \ell : \mathcal{C}_P \times (\mathcal{X} \times \mathcal{Y} \times \mathcal{A})^n \to \mathbb{R}$ and $g \circ \ell : \mathcal{C}_P \times (\mathcal{X} \times \mathcal{Y} \times \mathcal{A})^n \to \mathbb{R}$. To minimize $f \circ \ell$ subject to a constraint on $g \circ \ell$, we can for any $\theta \in \mathcal{C}_P$, and $G > 0$, a Lagrange multiplier, define the Lagrangian: $h(\theta, Z) = f \circ \ell(\theta, Z) + G \cdot \max(0, g \circ \ell(\theta, Z))$. The function $h$ has gradient

$$\nabla h(\theta, Z) = \nabla \ell(\theta, Z) \nabla f(\ell(\theta, Z)) + \tag{1}$$

$$G \cdot \mathbb{1}[g(\ell(\theta, Z)) \geq 0] \nabla \ell(\theta, Z) \nabla g(\ell(\theta, Z)) \tag{2}$$

where $\nabla \ell(\theta, Z)$ is a $P \times K$ matrix of the gradient of $\ell(\theta, Z)$ (or the transpose of its Jacobian), and $\nabla g(\ell(\theta, Z))$ is a $K \times 1$ column vector representing the gradient of $g$.

The Langrangian incorporates the constraint function $g$ while minimizing $f$. Following Agarwal et al. (2019), the Lagrangian for linear regression with BGL (definition 2.2) can be stated as follows:

$$\min_{\theta \in \mathcal{C}_P} \left[ \frac{1}{n} \sum_{i=1}^n (y_i - \theta(x_i))^2 \left( 1 + \sum_a \frac{\lambda_a}{n_a} \mathbb{1}[A_i = a] \right) \right]. \tag{3}$$

Solving the formulation above is challenging because there is no efficient algorithm for selecting the Lagrange multipliers in the general case. Agarwal et al. (2019) give a two-player game that eventually converges to an equilibrium. However, this game is not guaranteed to converge in polynomial time. Consequently, we use the constrained group objective optimization (CGOO) formulation to sidestep this issue. We now define the CGOO problem.

**Definition 3.2** (Constrained Group-Objective Optimization: $\mathrm{CGOO}(n, K, f, g, \ell, Z, \alpha)$). *Let $f : [0,1]^K \to \mathbb{R}$ be a function we wish to minimize subject to a constraint function $g : [0,1]^K \to \mathbb{R}$. Specifically, for any excess risk parameter $\alpha > 0$, decision set $\mathcal{C}$, and any dataset, $Z$, we wish to obtain a decision $\hat{\theta} \in \mathcal{C}$ such that*

$$f(\ell(\hat{\theta}, Z)) \leq \min_{\theta \in \mathcal{C}: g(\ell(\theta, Z)) \leq 0} f(\ell(\theta, Z)) + \alpha, \quad g(\ell(\hat{\theta}, Z)) \leq \alpha. \tag{4}$$

*Any procedure that takes input $Z$ and returns a decision $\hat{\theta} \in \mathcal{C}$ that satisfies Equation 4 is a Constrained Group-Objective optimization algorithm that solves the problem specified by $\mathrm{CGOO}(n, K, f, g, \ell, Z, \alpha) = \hat{\theta}$.*

---

**Algorithm 1** Stage 1: $\tau$-zCDP Standard Errors Computation for All Groups in Linear Regression

1: **Input**: $Z \in (\mathcal{X} \times \mathcal{Y} \times \mathcal{A})^n, K, \Delta, \tau > 0$
2: Split dataset $Z = (X, Y, A)$ into $Z_1, \ldots, Z_K$ for $K = |\mathcal{A}|$ groups as follows:
3: $X_1, X_2, \ldots, X_K \in \mathbb{R}^{n \times d}$
4: $Y_1, Y_2, \ldots, Y_K \in \mathbb{R}^n$
5: $n_1, n_2, \ldots, n_K$ integers
6:
7: $\tau_k = \tau/(K+1)$
8: Use budget of $\tau_k/2$ to sample $\widetilde{X^T X}$, a differentially private estimate of $X^T X$
9: Use budget of $\tau_k/2$ to sample $\widetilde{X^T Y}$, a differentially private estimate of $X^T Y$
10:
11: **if** $\lambda_{min}(\widetilde{X^T X}) \leq 0$ **then**
12:     // Returns "no information" or alternatively, halt/terminate
13:     **return** $1/\sqrt{K} \cdot \mathbf{1}_K$
14: **end**
15: // Calculate DP estimate of regression coefficients on entire dataset
16: $\tilde{\beta} = (\widetilde{X^T X})^{-1} \widetilde{X^T Y}$
17: **for** $k = 1$ **to** $k = K$ **do**
18:     $S_k = \left[ \sum_{i=1}^{n_k} (Y_{ki} - \langle X_{ki}, \tilde{\beta} \rangle)^2 |_0^{\Delta^2} + \mathcal{N}(0, \Delta^4/(2\tau_k)) \right]$
19:     **if** $S_k \leq 0$ **then**
20:         **return** $1/\sqrt{K} \cdot \mathbf{1}_K$
21:     **end**
22:     $s_k = \sqrt{\frac{S_k}{n_k}}$
23: **end for**
24: Normalize $s_1, s_2, \ldots, s_K$ so that $\sum s_k^2 = 1$
25: **return** $s_1, s_2, \ldots, s_K$

---

In the Lagrangian setup, if $f$, and $g$ are convex and Lipschitz, we can apply convex optimization algorithms that will result in a solution close to the global optima in time polynomial in $n, P, K$. Given these properties, we can solve the optimization problem using modified versions of (stochastic) gradient descent algorithms.

## 4 Group-aware DP Fair Regression

We now present our end-to-end differentially private two-stage algorithm for linear regression. The algorithms we present also directly address the Lagrangian fair regression problem discussed in Section 3. Even though we focus on linear regression in this work, our privacy allocation strategy is a meta-strategy that can be used for other estimators.

In the first stage, we estimate a precursor differentially private regression model on the data. We then calculate the standard error of the predictions for this regression model across all subgroups in the data. Given group standard error estimates, we allocate the privacy budget to prioritize groups with larger standard error of the prediction—i.e., more privacy budget is allocated to such groups. In the second stage, we solve the constrained linear regression formulation described in Section 3.2. Assuming that the gradient calls take constant time, it is easy to see that Algorithm 1 and Algorithm 2 runs in time $O(K \max_{k=1}^K n_k)$. In Algorithm 1, by using a privacy budget to sample a matrix, we mean that we compute the matrix in a differentially private manner (e.g., via the use of the exponential or gaussian mechanism). Also, we assume that the groups can overlap so we have to split the privacy budget for the $K$ groups. If there is no overlap, we can apply parallel composition— without splitting the privacy budget (Kairouz et al., 2017).

**Setup & Notation.** In Algorithms 1 and 2, for any $d \geq 1$, $\mathbf{1}_d$ is the all-ones $d$-dimensional vector and $I_{d \times d}$ is the identity diagonal $d \times d$ matrix. We rely on standard methods (e.g., see (Dwork et al., 2014)) to obtain differentially private estimates of $\widetilde{X^T X}, \widetilde{X^T Y}$, which are the variance and covariance matrices respectively

---

**Algorithm 2** Stage 2: $\mu$-zCDP Gradient Descent for Langrangian Fair Regression

---

1: **Input**: $Z \in (\mathcal{X} \times \mathcal{Y} \times \mathcal{A})^n$, $\eta_t$, $\Delta$, $T \geq 1$, $\mu > 0$, $G > 0$
2: Obtain DP standard errors $s_1, s_2, \ldots, s_K$ from Algorithm 1
3: Like in Algorithm 1, split dataset $Z = (X, Y, A)$ into $Z_1, \ldots, Z_K$ for $K = |\mathcal{A}|$ groups
4: Arbitrarily set $\theta_1 \in \mathcal{C}_P$
5: **for** $t = 1$ **to** $t = T$ **do**
6:     **for** $k = 1$ **to** $k = K$ **do**
7:         $\nabla \ell_k(\theta_t, Z) = \sum_{i=1}^{n_k} 2(\langle \theta_t, X_{ki} \rangle - Y_{ki}) X_{ki} = \sum_{i=1}^{n_k} \nabla \ell_k(\theta_t, X_{ki}, Y_{ki})$
8:         $\ell_k(\theta_t, Z) = \sum_{i=1}^{n_k} (\langle \theta_t, X_{ki} \rangle - Y_{ki})^2 = \sum_{i=1}^{n_k} \ell_k(\theta_t, X_{ki}, Y_{ki})$
9:         Use budget of $\mu_T = \mu \cdot s_k^2 / (2T)$ to sample $M_1^k$ as follows:
10:        $\nabla L_k(\theta_t, Z) = \frac{1}{n_k} \sum_{i=1}^{n_k} \nabla \ell_k(\theta, x_i, y_i)|^\Delta$ ($\ell_2$-norm per-example gradient clipping)
11:        $M_1^k = \nabla L_k(c, Z) + \mathcal{N}(0, I_{P \times P} \cdot \frac{\Delta^2}{2\mu_T})$
12:        Use budget of $\mu_T = \mu \cdot s_k^2 / (2T)$ to sample $M_2^k$ as follows:
13:        $L_k(\theta, Z) = \frac{1}{n} \sum_{i=1}^{n} \ell_k(\theta, X_{ki}, Y_{ki})|_0^{\Delta^2}$
14:        $M_2^k = L_k(\theta, Z) + \mathcal{N}(0, \frac{\Delta^4}{2\mu_T})$
15:     **end for**
16:     $\nabla h(\theta_t, Z)$ using $M_1 = (M_1^1, \ldots, M_1^K) \in \mathbb{R}^{P \times K}$ and $M_2 = (M_2^1, \ldots, M_2^K)^T \in \mathbb{R}^{K \times 1}$ (via Equation 2) as follows:
17:     $\nabla h(\theta_t, Z) = M_1 \nabla f(M_2) + G \cdot \mathbb{1}[g(M_2) \geq 0] M_1 \nabla g(M_2)$
18:     $\theta_{t+1} = \Pi_{\mathcal{C}_P}(\theta_t - \eta_t \nabla h(\theta_t, Z))$
19: **end for**
20: **return** $\theta_t$

---

(see Lemma D.1 in the appendix for proof details and mechanism for exact computation). We use $|_0^\Delta$ to denote the clipping operator. For a scalar $v$, $v|_0^\Delta$ returns $v$ clipped to lie in the range $[0, \Delta]$. For a vector $V$, $V|^\Delta$ returns $V$ clipped such that its $\ell_2$-norm is at most $\Delta$. A standard way to perform this operation is to return $Z/\max(1, \frac{\|Z\|_2}{\Delta})$.

**Stage 1: Standard Error Estimation.** First, we estimate, privately, the standard error of the prediction for a single linear regression model fit on data for all groups. Intuitively, the coefficients for this model will have higher structural uncertainty and standard error of the prediction for the smaller groups in the data. We then partition the data into $K$ groups and estimate each group's standard error by computing $s_k = \|Y_k - X_k \tilde{\beta}_k\| / \sqrt{n_k - d}$ but with noise added to ensure zCDP for the 'global' regression model $\tilde{\beta}$. Following previous work, we also assume that the group sizes, $n_1, \ldots, n_K$, are public, but that the record-level data is not. We allocate a privacy budget of $\tau/K$ for each group's standard error computation and normalize so that their squares sum to 1.

**Lemma 4.1** (Lemma C.1). *For any $\tau > 0$, Algorithm 1 is $\tau$-zCDP.*

**Stage 2: DP-based optimization.** In this stage of the formulation we use the privacy budget allocation from the first stage to learn a single linear predictor that satisfies the CGOO formulation (Section 3) via a differential private gradient descent scheme. Regular linear regression corresponds to the CGOO formulation without any fairness (Bounded Group Loss) constraints. We refer readers to the appendix for a version of Algorithm 2 for linear regression without fairness constraints. In this setting, we perform budget allocation as follows: for any $k \in [K]$, allocate $\mu \cdot s_k^2$ budget to calculating $\ell_k(c, Z)$ and $\nabla \ell_k(c, Z)$. Recall that $\ell_k$ is the loss for group $k$ and $\nabla \ell_k(c, Z)$ is its gradient.

We implement DPGD following the algorithm of Bassily et al. (2014) specialized to the ordinary least squares objective. For any fixed privacy budget $\mu > 0$, we allocate $\mu/2$ to $\ell$ and $\mu/2$ to $\nabla \ell$. Algorithm 2 spells out the crux of our procedure, using learning rate $\eta_t$.

**Lemma 4.2** (Lemma C.2). *For any $\mu > 0$, Algorithm 2 is $\mu$-zCDP.* [1]

---

[1] Algorithm 2 uses Algorithm 1 as a sub-routine. Algorithm 1 has its own privacy budget.

Based on Lemmas 4.1 and 4.2 and by basic composition of differential privacy (e.g., see (Dwork et al., 2006)), the entire scheme (Algorithm 1 and 2) is $(\tau + \mu) = \rho$-zCDP.

# 5 Lower Bounds

For private constrained group-objective optimization, we now show excess risk bounds. Note that these results are equivalent to a lower bound on the sample complexity required to solve the problem since these bounds are a function of the dataset size $n$. Our lower bound results are specialized to the CGOO setting (see Definition 3.2). For the functions $f, g$ analyzed below, one can think of $f$ as a function that computes an average or total loss over the $K$ group losses and $g$ as a function that computes the maximum or tail loss over the $K$ groups.

We attack the following question: *over the randomness of any $(\epsilon, 0)$ or $(\epsilon, \delta)$-DP algorithm, for a fixed dataset $D$ of size $n$, what is a lower bound for the accuracy of the mechanism that solves the constrained group-objective optimization problem (Definition 3.2)?*

Assuming the dataset is also drawn from $B_2^K$, we show a lower bound on the excess risk for decision set $\mathcal{C}_P = B_2^K = \{c \in \mathbb{R}^K : \|c\|_2 = 1\}$. That is, we consider the case where the decisions and datasets lie in the unit ball with $\ell_2$ norm. We show that for all $n, K \in \mathbb{N}$ and $\epsilon > 0$ there exists a dataset $D = \{x_i\}_{i=1}^n \subseteq B_2^K$ for which there is a CGOO problem with functions $f, g$ such that both $f$ and $g$ will have excess risk lower bounds of $\alpha \geq \Omega(\frac{K}{\epsilon n})$ and $\alpha \geq \Omega(\frac{\sqrt{K}}{\epsilon n})$ for any $(\epsilon, 0)$, $(\epsilon, \delta)$-differentially private algorithms respectively.

## 5.1 $(\epsilon, 0)$ Lower Bound

The major idea in the proof of Theorem 5.1 is to reduce to the problem of optimizing 1-way marginals (a standard method for lower bounding the accuracy of differentially private mechanisms).

**Theorem 5.1.** *Let $n, K \in \mathbb{N}, \epsilon > 0$ and $\alpha \in (0, 1]$. For every $\epsilon$-differentially private algorithm $\mathcal{M}$ that produces a decision $\hat{c} \in \mathcal{C}_P$ such that*

$$f(\ell(\hat{c}, D)) \leq \min_{c \in \mathcal{C}_P : g(\ell(c,D)) \leq 0} f(\ell(c, D)) + \alpha, \qquad g(\ell(\hat{c}, D)) \leq \alpha,$$

*there is a dataset $D = \{\boldsymbol{x}_1, \ldots, \boldsymbol{x}_n\} \subseteq B_2^K$ such that, with probability at least $1/2$, we must have $\alpha \geq \Omega\left(\frac{K}{\epsilon n}\right)$ (or equivalently, $n \geq \Omega\left(\frac{K}{\epsilon \alpha}\right)$) where $f, g$ are Lipschitz, smooth functions defined as follows:*

$$f(\ell(c, D)) = -\frac{1}{n} \sum_{i=1}^n \langle c, \boldsymbol{x}_i \rangle, \qquad g(\ell(c, D)) = f(\ell(c, D)) + \frac{1}{n} \|\sum_{i=1}^n \boldsymbol{x}_i\|.$$

*for all $c \in B_2^K$.*

*Proof.* We have defined $f$ as $f(\ell(c, D)) = -\frac{1}{n} \sum_{i=1}^n \langle c, \mathbf{x}_i \rangle$ which has minimum $c^* = \frac{\sum_{i=1}^n \mathbf{x}_i}{\|\sum_{i=1}^n \mathbf{x}_i\|}$ by Lemma A.1. We defined $g$ as $g(\ell(c, D)) = f(\ell(c, D)) + \frac{1}{n} \|\sum_{i=1}^n \mathbf{x}_i\| = f(\ell(c, D)) - f(\ell(c^*, D))$ which has minimum $c^*$ so that the constraint $g(\ell(c^*, D)) \leq 0$ is satisfied.

Now by Lemma A.2, we have that $f(\ell(c, D)) - f(\ell(c^*, D)) = \frac{\|\sum_{i=1}^n \mathbf{x}_i\|}{2n} \|c - c^*\|^2$. Now we invoke Lemma A.3. If $\hat{c}$ is the output of any $\epsilon$-differentially private mechanism $\mathcal{M}$ then we must have that $\|c - c^*\| = \Omega(1)$. Suppose not. Then that would imply that we can construct a new mechanism $\mathcal{M}'$ that outputs $\hat{c} \cdot \frac{\|\sum_{i=1}^n \mathbf{x}_i\|}{n}$ which would contradict Lemma A.3. As a result, $\|c - c^*\| = \Omega(1)$ so that $f(\ell(\hat{c}, D)) - f(\ell(c^*, D)) = \Omega(\frac{K}{\epsilon n})$ for the output $\hat{c}$ of any $\epsilon$ differentially private mechanism. $\square$

## 5.2 $(\epsilon, \delta)$ Lower Bound

For the proof of Theorem 5.2, we follow the steps of the proof for Theorem 5.1 but invoke the lower bound for 1-way marginals in the approximate DP case (and not the pure case).

**Theorem 5.2.** *Let $n, K \in \mathbb{N}, \epsilon > 0, \alpha \in (0, 1]$, and $\delta = o(\frac{1}{n})$. For every $(\epsilon, \delta)$-differentially private algorithm $\mathcal{M}$ that produces a decision $\hat{c} \in \mathcal{C}_P$ such that*

$$f(\ell(\hat{c}, D)) \leq \min_{c \in \mathcal{C}_P : g(\ell(c,D)) \leq 0} f(\ell(c, D)) + \alpha, \qquad g(\ell(\hat{c}, D)) \leq \alpha,$$

*there is a dataset $D = \{\boldsymbol{x}_1, \ldots, \boldsymbol{x}_n\} \subseteq B_2^K$ such that, with probability at least 1/3, we must have $\alpha \geq \Omega\left(\frac{\sqrt{K}}{\epsilon n}\right)$ (or equivalently, $n \geq \Omega\left(\frac{\sqrt{K}}{\epsilon \alpha}\right)$) where $f, g$ are Lipschitz, smooth functions defined as follows:*

$$f(\ell(c, D)) = -\frac{1}{n}\sum_{i=1}^{n}\langle c, \boldsymbol{x}_i \rangle, \qquad g(\ell(c, D)) = f(\ell(c, D)) + \frac{1}{n}\|\sum_{i=1}^{n}\boldsymbol{x}_i\|.$$

*for all $c \in B_2^K$.*

*Proof.* Again, the way we have defined $f$, by Lemma A.2, we have that $f(\ell(c, D))) - f(\ell(c^*, D)) = \frac{\|\sum_{i=1}^{n}\mathbf{x}_i\|}{2n}\|c - c^*\|^2$. Now we invoke Lemma A.4.

If $\hat{c}$ is the output of any $(\epsilon, \delta)$-differentially private mechanism $\mathcal{M}$ then we must have that $\|c - c^*\| = \Omega(1)$. Suppose not. Then that would imply that we can construct a new mechanism $\mathcal{M}'$ that outputs $\hat{c} \cdot \frac{\|\sum_{i=1}^{n}\mathbf{x}_i\|}{n}$ which would contradict Lemma A.4. As a result, $\|c - c^*\| = \Omega(1)$ so that $f(\ell(\hat{c}, D)) - f(\ell(c^*, D)) = \Omega(\frac{\sqrt{K}}{\epsilon n})$ for the output $\hat{c}$ of any $(\epsilon, \delta)$-differentially private mechanism. $\qquad\square$

## 6 Empirical Evaluation

We now discuss the empirical performance of our proposed scheme for synthetic and real-world datasets. We discuss the results for the law school dataset in the appendix. We find that the proposed group-aware privacy budget allocation leads to 10% to 50% reduction in mean standard error of the prediction for the smallest groups. For real-world datasets, we find even more substantial gains. For all our experiments, the total privacy budget is $\rho = \tau + \mu$. We allocate 20% of the budget to stage 1 (i.e., $\tau = 0.2\rho$) and 80% to stage 2 (i.e., $\mu = 0.8\rho$). We present results for other budget allocation schemes between the two stages in the Appendix. In the results, 'DPSE' refers to our proposed scheme and 'DPNoSE' refers to standard linear regression computed via DPGD (budget of $\rho$) but with no group aware budget allocation.

We also tried our methods on data with more than 2 groups (in particular, $K = 5$ and $K = 10$ groups). For simplicity, we focus on the $K = 2$ case (one majority group and one minority group) since it is already very nuanced in this case.

In this section, we present a subset of our empirical results. More experimental details can be found in the appendix. See Section B.

**Setup for Synthetic Experiments.** For the results presented here, we assume that $\mathcal{C}_P = [-5, 5]^P$. We define $\ell$ to be a multivariate function such that for any group $k \in [K]$ and dataset $Z$ with $n_k$ items with features $X_{k1}, X_{k2}, \ldots, X_{kn_k} \in \mathbb{R}^P$. Let $\ell_k(c, Z) = \sum_{i=1}^{n_k}(Y_i - \langle X_{ki}, c \rangle)^2$. Its gradient is $\nabla \ell_k(c, Z)$. We set $f(l) = \sum_{i=1}^{K} l_i$ and $g(l) = \max_{i \in [K]} l_i$. For the synthetic data experiments, we assume the following ordinary least squares (OLS) generation model: for any group $k \in [K]$ and for all $i \in [n_k]$, $Y_i = X_{ki}^T \beta_k + \mathcal{N}(0, I\sigma_k^2)$ where $\beta_k \in \mathbb{R}^P$, $\sigma_k > 0$, $X_{ki} \in \mathbb{R}^P$.

**Overview of Results: Synthetic Datasets.** Figure 6 shows different parameter settings for a two-group synthetic scenario where we observe what happens to the MSPE as we vary the privacy parameter $\rho$. The blue line shows the MSPE for the smallest subgroup when the privacy budget is set with our group-aware scheme while the cyan line is the MSPE for the smallest group when the privacy budget is not tailored to any group. For reasonable levels of privacy, we observe substantial (10% to 50%) reduction in the MSPE when we set the privacy budget using our proposed formulation. When we incorporate fairness constraints, we find

that the group aware formulation (DPSE smaller group) still maintains the 10% to 50% reduction in the MSPE for the smaller groups.

$K > 2$ **Groups and Group Sizes.** We also considered additional settings with more than 2 groups. In this case, we still observe at least a 10 percent reduction in MSPE for the smallest groups without hurting the larger groups. A concern might be that as $K$ (the number of groups) gets larger, the standard error calculation in stage 1 becomes more inaccurate. In this case, one could allocate more privacy budget to stage 1. We conducted comprehensive experiments for $K = 2, 5, 10, 15, 20$ groups. Similarly, we include additional discussion empirical results on the law school and credit datasets in the Appendix of the original paper. For the sake of brevity, we present tables here for the $K = 5$ case. In the first experiment, the largest group (group 1) has size 10,000 and the smallest has size 500 (so a 20x ratio). In the second experiment, the largest group has size 5,000 and the smallest has size 100 (so a 50x ratio). For all the other cases ($K = 10, 15, 20$), we observe similar gains in Mean Squared Prediction Error (MSPE).

**Logistic Regression.** Our methods can be *easily* translated to the logistic regression (for use in classification) setting. For training data $Z = \{(\mathbf{x}_i, y_i)\}_{i=1}^n$, the linear regression loss is

$$h(\theta \mid Z) = \frac{1}{n} \sum_{i=1}^n (\langle \theta, x_i \rangle - y_i)^2,$$

which has an explicit gradient. We can replace this with the logistic loss as follows

$$h(\theta \mid Z) = \frac{1}{n} \sum_{i=1}^n \log(1 + e^{-y_i \theta^T x_i}),$$

where $y_i$ are the labels in this case. The logistic loss also has an explicit gradient. And since our algorithms require only oracle access to the gradients, the change to compute the logistic regression results is minimal.

| Group | DPNoSE ($\rho = 2$) | DPSE ($\rho = 2$) | Group | DPNoSE ($\rho = 4.5$) | DPSE ($\rho = 4.5$) |
|---|---|---|---|---|---|
| 1 | 1.02 | 1.27 | 1 | 1.01 | 1.26 |
| 2 | 18.71 | 15.75 | 2 | 18.69 | 15.73 |
| 3 | 21.09 | 17.75 | 3 | 21.07 | 17.73 |
| 4 | 21.35 | 17.94 | 4 | 21.34 | 17.93 |
| 5 | 18.11 | 15.29 | 5 | 18.09 | 15.28 |

| Group | DPNoSE ($\rho = 8$) | DPSE ($\rho = 8$) |
|---|---|---|
| 1 | 1.01 | 1.25 |
| 2 | 18.64 | 15.71 |
| 3 | 21.02 | 17.71 |
| 4 | 21.29 | 17.91 |
| 5 | 18.05 | 15.26 |

Table 1: MSPE for DPSE vs. DPNoSE. (5 groups, 20x)

## 6.1 Evaluating Different Privacy Splits Based on Dataset Characteristics

In the main paper, our methodology is such that we can specify any privacy budget splits between the 2 stages via $(\tau, \mu)$ as long as $\rho = \tau + \mu$. However, in our experimental evaluation, we set $\tau = 0.2\rho$. Note that the gains of our approach will likely be bigger when the standard errors are larger in the minority group. Now we show, via some experimental validation, that using 20% of the privacy budget allows us to privately estimate the standard errors in the first stage that we can then use for privacy budget allocation. For simplicity, we only generate 2 groups. $n_1, n_2$ are the sizes of the majority and minority groups, respectively. We see from Table 2 that since the minority group is much smaller in size than the majority group, it tends to have a much larger relative standard error. As a result, majority of the privacy budget is used for privately estimating quantities related to the minority group. See Tables 2 and 3.

| $\rho$ | Standard Errors (Majority-Minority) | Normalized Budget Splits (Majority-Minority) |
|---|---|---|
| $2^2/2$ | [0.00043513 0.12542164] | [1.20362660e-05 9.99987964e-01] |
| $3^2/2$ | [0.00078679 0.04677319] | [2.82876770e-04 9.99717123e-01] |
| $4^2/2$ | [0.00040894 0.03935408] | [1.07969485e-04 9.99892031e-01] |
| $5^2/2$ | [0.00038963 0.03800898] | [1.05070219e-04 9.99894930e-01] |

Table 2: Standard Errors and Splits. $n_1 = 8000000, n_2 = 500$

| $\rho$ | Standard Errors (Majority-Minority) | Normalized Budget Splits (Majority-Minority) |
|---|---|---|
| $3^2/2$ | [0.00053533 0.05429162] | [9.72139296e-05 9.99902786e-01] |
| $4^2/2$ | [0.00018697 0.03429942] | [2.97124804e-05 9.99970288e-01] |
| $5^2/2$ | [0.00015344 0.04948024] | [9.61694723e-06 9.99990383e-01] |

Table 3: Standard Errors and Splits. $n_1 = 16000000, n_2 = 500$

**Why Even Look at Standard Errors?**  We have now added additional experiments to illustrate why looking at the standard errors could provide information that is **impossible** to obtain by just looking at the data set sizes. In particular, we provide datasets where two groups have **exactly** the same data set sizes. So just looking at the sizes alone would result in the same privacy split. However, one group has much larger sampling error and the other does not. According to our methodology, more budget would be allocated to the group with larger sampling error.

We vary the variance in the independent variable as follows: the first group (group 1) has variance 10000 and the second (group 2) has variance of 10 (much smaller!). To account for the severe imbalance in the variance, our methodology assigns a privacy budget split that favors the group with much larger variance. We show these results in Table 4.

## 7   Additional Related Work

**The Disparate Impact Due to Differential Privacy & Langrangian Approaches.** Our work was inspired by the findings of Bagdasaryan et al. (2019), who showed that predictors learned via DPSGD have low performance for minority groups. Similarly, Pujol et al. (2020) show, in a resource allocation setting, that under $\epsilon$-differential privacy, the noise added disproportionately affects minority groups in the data.

Mahdi Khalili et al. (2020) provide an approach that uses the exponential mechanism as a post-processing step to improve the downstream fairness and privacy properties of standard supervised learning model.Tran et al. (2020) address satisfying privacy and fairness constraints through a Langrangian duality approach. We also rely on a Langrangian framework to incorporate fairness constraints in the setting we consider. However, we focus on bounded group loss definitions of fairness in our formulation, which Tran et al. (2020) does not consider. Using Lagrangian duality for privacy (via linear programming, for example) has been considered before, in different contexts, in the differential privacy literature (Nikolov et al., 2013; Hsu et al., 2014).

More recently, Xu et al. (2019); Esipova et al. (2023) show results for achieving differential privacy and a range of notions of algorithmic fairness (Hardt et al., 2016). These impossibility results do not apply to bounded group loss, which is the notion of disparity that we consider in this work. Mozannar et al. (2020) consider the task of learning fair predictors without access to sensitive attributes under local differential privacy.

| $n$ | Standard Errors (Majority-Minority) | Normalized Budget Splits (Majority-Minority) |
|---|---|---|
| $n = 1000$ | [4.04838336, 96.67176174] | [0.04019438, 0.95980562] |
| $n = 50000$ | [3.18102322, 100.06187079] | [0.03081106, 0.96918894] |
| $n = 9000000$ | [3.16276276, 100.01852322] | [0.03065248, 0.96934752] |

Table 4: Standard Errors and Splits. $n = 1000, 50000, 9000000$.

**Algorithmic Fairness.** Predictors learned via empirical risk minimization (ERM) often exhibit poor performance for minority, both in size and by category, groups in the data (Buolamwini & Gebru, 2018; Koenecke et al., 2020; Barocas et al., 2019). In this work, we also consider learning regression models subject to fairness constraints using a Lagrangian formulation that builds on the work of Agarwal et al. (2019). Khalili et al. (2021); Tran et al. (2021b). Agarwal et al. (2019) present algorithms to obtain linear regression predictors subject to statistical parity and bounded sub-group loss; specifically, they study how to transform a fair regression problem with constraints to a generic loss minimization problem, via Lagrange duality, without constraints. We abstract the fair constrained regression problem into one that we term a "constrained group-objective optimization" problem, where the goal is to minimize an objective with certain properties (say Lipschitzness and/or Convexity), but for which there are 'group' constraints. The Lagrangian fair linear regression formulation that we consider falls under their abstraction.

**Group-Aware Learning.** The principle underlying the group-aware scheme that we propose requires *treating* different subgroups in the data *differently*. For example, Hébert-Johnson et al. (2018) proposed multicalibration, a principle that requires calibrated predictions for all subgroups in the training data that can be identified. Building on this work, Kim et al. (2019) propose *multiaccuracy boost*, an approach, using group reweighting, to audit and improve the performance of the predictive model among subgroups in the data that have low performance. Alternatively, one can also learn group-specific predictors for each subgroup in the data (Dwork et al., 2018). Koyejo et al. (2014) and Hiranandani et al. (2020) explore the goal of learning functions of partitions of the confusion matrix with applications to algorithmic fairness.

## 8 Conclusion

In this paper, we proposed a group-aware approach to set the privacy budget while learning regression models for datasets with stratification. This proposed formulation seeks to overcome the disparate model performance that naive application of differential privacy incurs for minority subgroups. We propose to allocate the privacy budget for each subgroup of the dataset based on the standard error of a precursor regression model. This formulation allows us to reduce the level of noise that DP adds to groups where the parameter estimation will be most uncertain—usually groups with smaller sizes. We justify the group-aware privacy budget allocation with a theoretical analysis of an idealized 2-mixture of Gaussians model. Finally, we show that the proposed scheme leads to a reduction in the mean square prediction error, compared to the standard privacy formulation, on synthetic and real-world datasets where we observe substantial (10% to 50%) reduction in the MSPE when we set the privacy budget using our proposed formulation.

We however note that it is unclear whether our current proposal extends directly to deep learning and modern neural networks. We believe this challenge presents avenue for intriguing future work. There are straightforward modifications to our scheme that would extend to the deep learning setting.

Our proposed group-aware privacy budget allocation shows promise; however, there still remain open questions. As it stands, it is unclear how to, optimally, split the overall privacy budget between the two stages of the algorithm. We make this decision a user-dependent one, but future work will look at how to determine the privacy split in an optimal fashion via analysis of the Pareto frontier of the utility-privacy trade-off. We considered the bounded subgroup loss formulation here, however, future work can also consider more sophisticated fairness constraints.

Our framework is centered on linear regression, a fundamental task; however, extensions to other estimators including those with non-convex loss functions is also desirable. Taken together, we hope this work opens up discussion about approaches for mitigating the disparate impact of differential privacy for small subgroups.

### Acknowledgments

Daniel Alabi was supported by the Simons Foundation (965342, D.A.) as part of the Junior Fellowship from the Simons Society of Fellows and a Fellowship from Meta AI during his graduate studies at Harvard University. We thank Julius Adebayo for his contributions to early stages of this project.

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

# A    Helper Lemmas for Section 5

**Lemma A.1.** *Let $c^* = \arg\min_{c:\|c\|\geq 1} -\frac{1}{n}\sum_{i=1}^n \langle c, \boldsymbol{x}_i\rangle$ where $\boldsymbol{x}_i \in B_2^K$ for all $i \in [n]$, then $c^* = \frac{\sum_{i=1}^n \boldsymbol{x}_i}{\|\sum_{i=1}^n \boldsymbol{x}_i\|}$.*

*Proof.* For any $c \in B_2^K$ we have $|\langle c, \sum_{i=1}^n \mathbf{x}_i\rangle| \leq \|c\|\|\sum_{i=1}^n \mathbf{x}_i\| = \|\sum_{i=1}^n \mathbf{x}_i\|$ by Cauchy-Schwarz and this is tight when $c = \frac{\sum_{i=1}^n \mathbf{x}_i}{\|\sum_{i=1}^n \mathbf{x}_i\|}$ or $c = -\frac{\sum_{i=1}^n \mathbf{x}_i}{\|\sum_{i=1}^n \mathbf{x}_i\|}$. As a result, the minimum of $-\frac{1}{n}\sum_{i=1}^n \langle c, \mathbf{x}_i\rangle$ is attained at $c = \frac{\sum_{i=1}^n \mathbf{x}_i}{\|\sum_{i=1}^n \mathbf{x}_i\|}$. $\qquad\square$

**Lemma A.2.** *Let $f(\ell(c, D)) = -\frac{1}{n}\sum_{i=1}^n \langle c, \boldsymbol{x}_i\rangle$ where $c, \boldsymbol{x}_i \in B_2^K$ for all $i \in [n]$, then*

$$f(\ell(c, D)) - f(\ell(c^*, D)) = \frac{\|\sum_{i=1}^n \boldsymbol{x}_i\|}{2n}\|c - c^*\|^2$$

*for any $c \in B_2^K$ and $c^* = \frac{\sum_{i=1}^n \boldsymbol{x}_i}{\|\sum_{i=1}^n \boldsymbol{x}_i\|}$.*

*Proof.* We have that

$$f(\ell(c, D)) - f(\ell(c^*, D)) = \frac{1}{n}\sum_{i=1}^n (\langle c^*, \mathbf{x}_i\rangle - \langle c, \mathbf{x}_i\rangle) \tag{5}$$

$$= \frac{1}{n}\left(\langle c^*, \sum_{i=1}^n \mathbf{x}_i\rangle - \langle c, \sum_{i=1}^n \mathbf{x}_i\rangle\right) \tag{6}$$

$$= \frac{1}{n}\left(\|\sum_{i=1}^n \mathbf{x}_i\| - \langle c, \sum_{i=1}^n \mathbf{x}_i\rangle\right) \tag{7}$$

$$= \frac{\|\sum_{i=1}^n \mathbf{x}_i\|}{n}\left(1 - \langle c, c^*\rangle\right) \tag{8}$$

$$= \frac{\|\sum_{i=1}^n \mathbf{x}_i\|}{2n}\|c - c^*\|^2 \tag{9}$$

where we have used that $\|c - c^*\|^2 = \|c\|^2 + \|c^*\|^2 - 2\langle c, c^*\rangle = 2 - 2\langle c, c^*\rangle$ and $c^* = \frac{\sum_{i=1}^n \mathbf{x}_i}{\|\sum_{i=1}^n \mathbf{x}_i\|}$. $\qquad\square$

We now state lower bound lemmas for 1-way marginals. Lemma A.3 shows the lower bound for 1-way marginals for $\epsilon$-differentially private algorithms and Lemma A.4 is for $(\epsilon, \delta)$-differentially private algorithms.

**Lemma A.3** (Part 1 of Lemma 5.1 in (Bassily et al., 2014))**.** *Let $n, K \in \mathbb{N}$ and $\epsilon > 0$. There exists a number $M = \Omega(\min(n, \frac{K}{\epsilon}))$ such that for every $\epsilon$-differentially private algorithm $\mathcal{M}$ there is a dataset $D = \{\boldsymbol{x}_1, \ldots, \boldsymbol{x}_n\} \subseteq B_2^K$ with $\|\sum_{i=1}^n \boldsymbol{x}_i\|_2 \in [M-1, M+1]$ such that, with probability at least 1/2 (over the randomness of the algorithm), we have*

$$\|\mathcal{M}(D) - q(D)\|_2 = \Omega(\min(1, \frac{K}{\epsilon n})),$$

*where $q(D) = \frac{1}{n}\sum_{i=1}^n \boldsymbol{x}_i$.*

**Lemma A.4** (Part 2 of Lemma 5.1 in (Bassily et al., 2014))**.** *Let $n, K \in \mathbb{N}$, $\epsilon > 0$, and $\delta = o(\frac{1}{n})$. There is a number $M = \Omega(\min(n, \frac{\sqrt{K}}{\epsilon}))$ such that for every $(\epsilon, \delta)$-differentially private algorithm $\mathcal{M}$, there is a dataset $D = \{\boldsymbol{x}_1, \ldots, \boldsymbol{x}_n\} \subseteq B_2^K$ with $\|\sum_{i=1}^n \boldsymbol{x}_i\|_2 \in [M-1, M+1]$ such that, with probability at least 1/3 (over the randomness of the algorithm), we have*

$$\|\mathcal{M}(D) - q(D)\|_2 = \Omega(\min(1, \frac{\sqrt{K}}{\epsilon n})),$$

*where $q(D) = \frac{1}{n}\sum_{i=1}^n \boldsymbol{x}_i$.*

# B  Additional Experimental Details

In this section, we present additional empirical evaluation of the our proposed group-aware differentially private budget allocation algorithm. For our experimental evaluation, the method that uses standard error information is called DPSE while the method that doesn't use such information is called DPNoSE.

**Code**    We include a copy of a Numpy implementation of the algorithms that we present in the main paper along with the a README and a requirements file.

**Trials & Compute Used**    Our results are computed over different instances of mixtures. Each run on a mixture is evaluated 1000 (random seeds) times and results are averaged over these Monte Carlo trials. We find that the average standard deviation of the Mean Squared Prediction Error (MSPE) of our results is less than 10% of the MSPE. All our experiments are run on a MacBook Pro (13-inch, 2018) with a 2.3GHz Quad-Core Intel Core i5 with 16GB Memory.

**Real-World Datasets**    In this section, we give a more detailed overview of the Credit dataset, which was discussed in the paper. In addition, we present empirical results on the Law School dataset.

**Law School Dataset Description**    The second real-world data set that we consider consists of demographic variables for law school students derived from the Longitudinal Bar Passage Study (Wightman, 1998). In this data set, there are attributes including the type of law school attended, the LSAT score, the undergraduate grade point average (GPA), the age, the race, the first year law school GPA, and the final cumulative GPA of each applicant. We consider the task of predicting the final cumulative GPA of each applicant given the underlying demographic variables. We take the attribute race as the sensitive attribute in the fair regression framework. We consider two groups: the 'white' students (22610 samples), and the 'black' students (1874 samples). We normalize all features to lie in an $\ell_2$ ball.

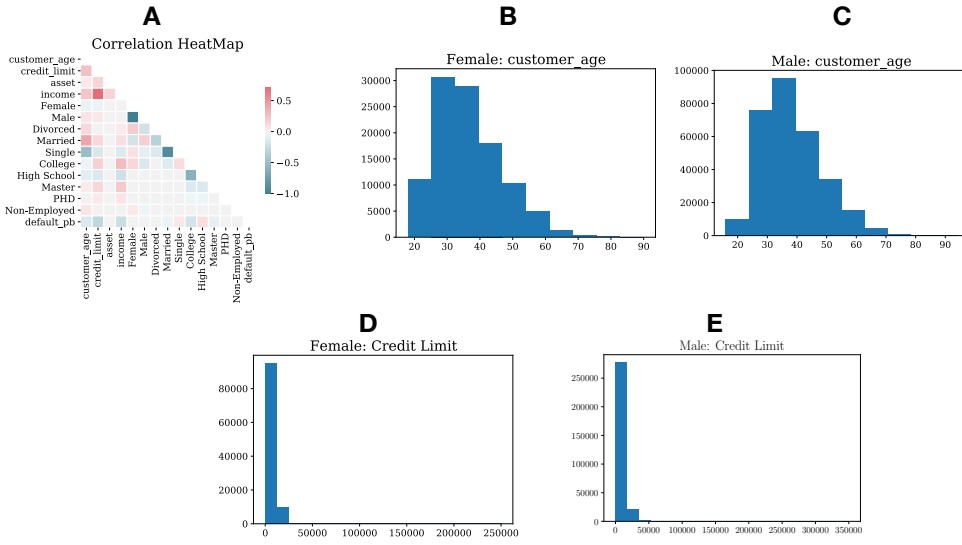

Figure 1: **Data set overview.** We examine some statistics of the Credit dataset. A) The variables present in the Credit dataset and the correlation heatmap amongst these variables. We see that asset, income, and educational status are the variables relatively more highly correlated with the credit limit than gender. B) The distribution of customer age for Females in the dataset. C) The distribution of customer age for Males in the dataset. D) The distribution of credit limit for females in the dataset. E) The distribution of credit limit for males in the dataset.

**Credit Dataset.**    First, as discussed in the paper, the key task in the credit setting is that of predicting the credit limit assigned to an individual given socio-economic variables like income, asset, and Age. In Figure 1,

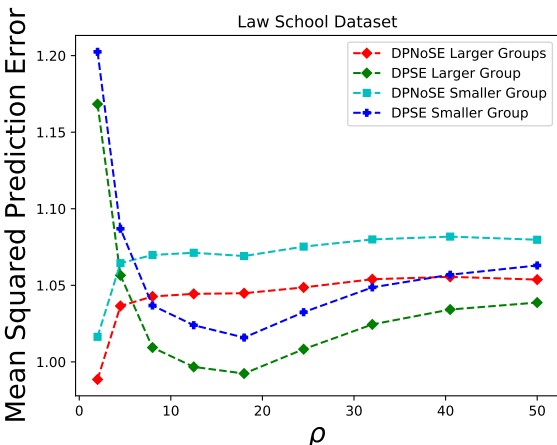

Figure 2: Empirical Results on the Law School Dataset.

we show a heat map of the correlation amongst these variables as well as histograms of age and credit limit for the male and female groups in the dataset. For the experiments performed, we subsampled the dataset into two: 11,480 males and 2,937 females.

In Figure 3, we show that DPSE outperforms DPNoSE on the smallest subgroup. Let $\beta_1, \beta_2$ denote the slopes in the 2 groups in the dataset. We see about a $> 50\%$ improvement in MSPE, on average, for slopes $(\beta_1, \beta_2) = (-10, 10)$ and $(\beta_1, \beta_2) = (10, -10)$ whether the ratio of sample sizes is 5x or 50x. The improvement becomes small when the ratio of the slopes is much higher (i.e., when $(\beta_1, \beta_2) = (-10, -1)$). We vary the privacy parameter from $\rho = 0.1^2$ to $\rho = 10^2/2$ and our Monte Carlo results are averaged over 1000 trials. $\sigma_e$ is the noise in the dependent variable for both groups. We set $\sigma_e = 1$. e.g., for group 1, $Y_1 \sim \mathcal{N}(X_1\beta_1, \sigma_e^2 I_{n_1 \times n_1})$, where $n_1$ is the number of data points in group 1, the majority group.

Figure 2 shows the empirical results for regression models trained on the law school data. Again, we see that the group-aware budget allocation approach (DPSE) provides a reduction in the mean squared prediction error of the resulting models. Taken together, these results further demonstrate that our proposed approach reduces the disparate impact of differential privacy in small groups.

### B.1 MSPE Results for DPSE & DPNoSE for additional synthetic settings.

In all the synthetic data experiments below, the clipping bound (for the gradients and losses) is set to $\Delta = 2$. We make this choice because given the synthetic setting that we consider, the Lipschitz constant of the is Lagrangian formulation function is *usually* at most 2. We note here that the clipping bound here is applied to the gradients and losses and not the OLS parameters learned.

**Beyond Linear Regression.** We replicate the MNIST "8-vs-2" binary classification experiment from Bagdasaryan et al. (2019). We use a logistic regression model in the first stage, and a LeNet network for the second stage (total $\epsilon = 5.73$). This means that we tailor the group privacy budgets based on the residuals of the logistic regression for the LeNet model. We find a 64 percent reduction in classification error, for the minority '8' class, compared to the results of Bagdasaryan et al. (2019). This experiment suggests that our approach can generalize to other model classes.

## C  Proofs and Additional Detail on Algorithms for Group-Aware OLS

**Lemma C.1.** *For any $\tau > 0$, Algorithm 1 is $\tau$-zCDP.*

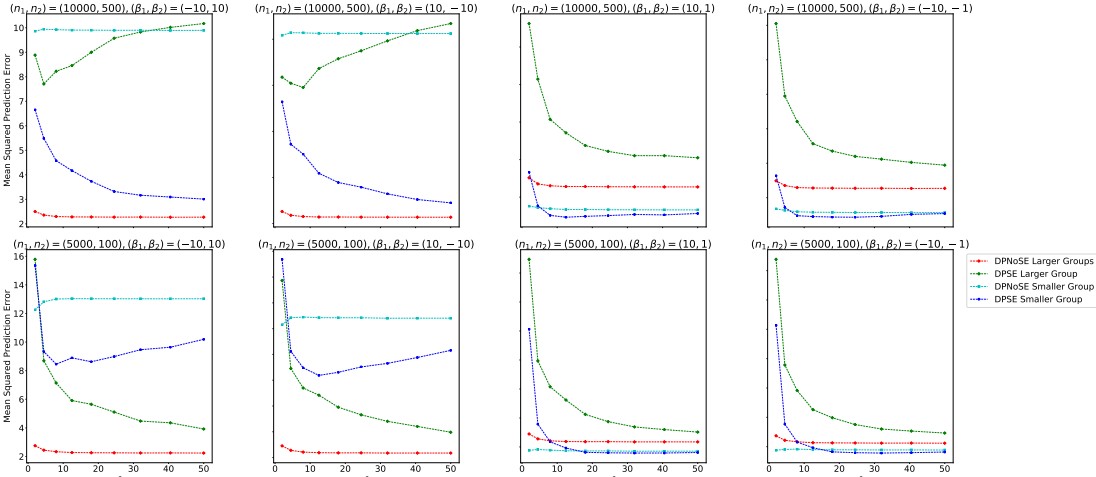

Figure 3: **Additional Empirical Results for Synthetic Datasets**. We vary the sample size ratio between the larger and smaller groups from 20x to 50x. We also vary the slopes used to generate the OLS models within each group. We consider group slopes $\{(-10, 10), (10, -10), (10, 1), (-10, -1)\}$. Across most regimes, we see that the MSPE in the smallest group is reduced with the DPSE procedure.

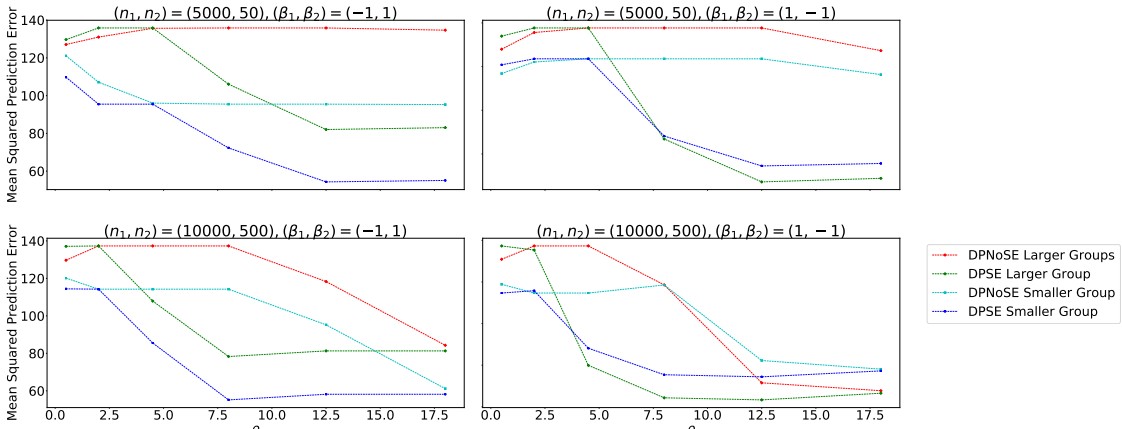

Figure 4: **Additional Empirical Results for Synthetic Datasets**. We vary the sample size ratio between the larger and smaller groups from 20x to 100x. We also vary the slopes used to generate the OLS models within each group. We consider group slopes $\{(-1, 1), (1, -1)\}$. We also vary the noise in the dependent variable and observe similar results. Across all regimes, we see that the MSPE in the smallest group is reduced with the DPSE procedure.

*Proof.* First, recall Lemma 2.3 in (Bun & Steinke, 2016) (for composition and postprocessing of zCDP) and Lemmas 2.4, 2.5 for the application of the Gaussian Mechanism for zCDP. We use $\tau/(K+1)$ to calculate DP estimates of the covariance and variance matrices: $\widetilde{X^T X}$ and $\widetilde{X^T Y}$. If the minimum eigenvalue of $\widetilde{X^T X}$ is $\leq 0$ (which implies noninvertibility or non-positive variances), we return a normalized all-ones vector, a step that is still private by postprocessing DP capabilities.

We proceed to calculate the DP estimate of the standard error (SE) for each group using Lemma 2.4 in (Bun & Steinke, 2016) to satisfy $\tau/(K+1)$-zCDP. Again, if the SE results in a value $\leq 0$, we return a normalized all-ones vector. Finally, by composition properties, the whole procedure satisfies $\tau$-zCDP. $\square$

**Lemma C.2.** *For any $\mu > 0$, Algorithm 2 is $\mu$-zCDP.*

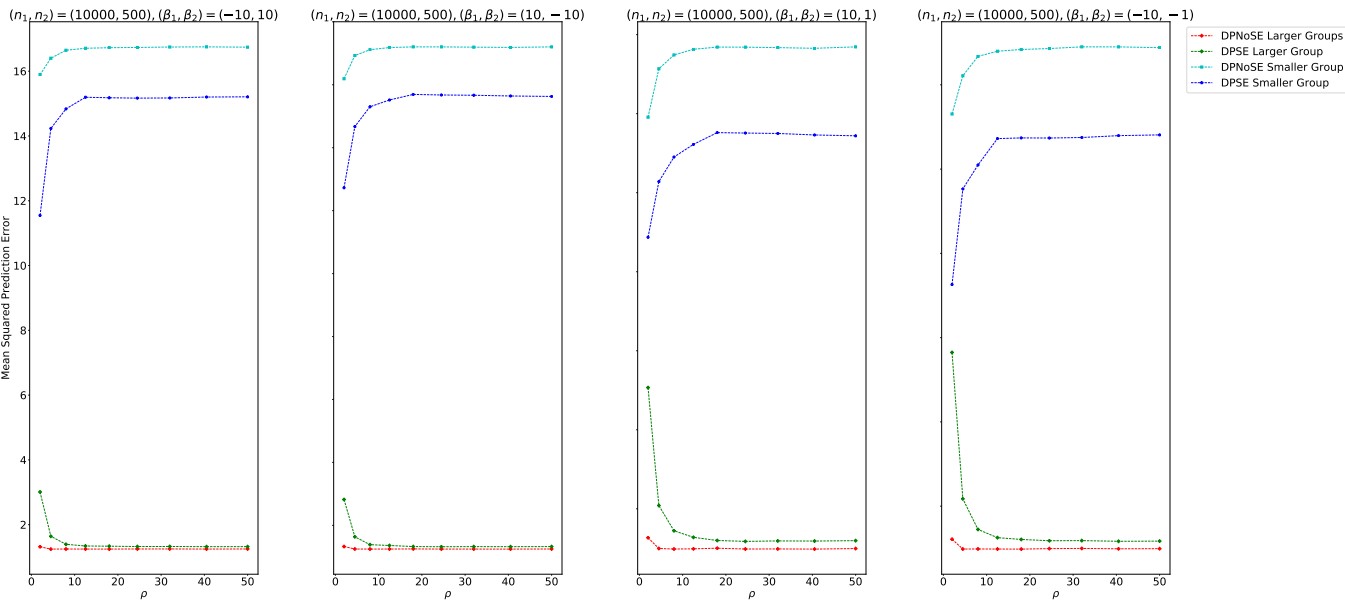

Figure 5: **Empirical Results for Synthetic Datasets**. We set the sample size ratio between the larger and smaller groups to 100x. We vary the slopes used to generate the OLS models within each group. We consider group slopes $\{(-10, 10), (10, -10), (10, 1), (-10, -1)\}$. Across most regimes, we see that the MSPE in the smallest group is reduced with the DPSE procedure. In this case, the loss in the larger group is even much smaller given the large size of the larger group.

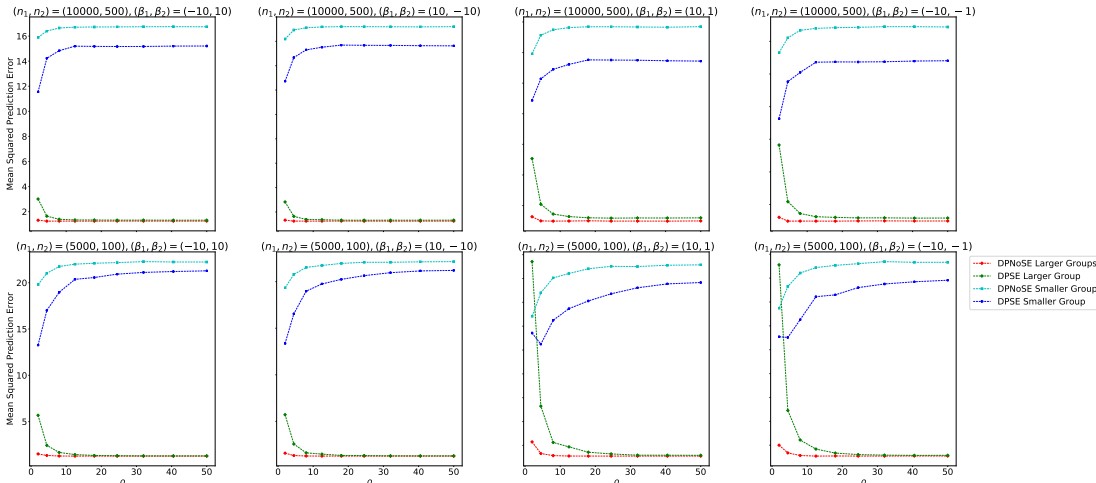

Figure 6: **Empirical Results for Synthetic Datasets**. We observe, on average, a 10-20% reduction in the MSPE. For small privacy parameters, we tend to overpenalize the larger group, leading to a quicker convergence to the right parameters for the smaller group.

*Proof.* Again, we apply Lemma 2.3 in (Bun & Steinke, 2016) (for composition and postprocessing of zCDP) and Lemmas 2.4, 2.5 for the Gaussian Mechanism zCDP application.

There are $T$ iterations so we split our privacy budget into $\mu/T$. Since from stage 1, $\sum s_k^2 = 1$, for each group $k \in [K]$, we use budget of $s_k^2 \cdot \mu/T$ (split into two for computing $\ell_k$ and its gradient $\nabla \ell_k$). By post-procesing and composition properties of zCDP, the whole procedure satisfies $\mu$-zCDP.

$\square$

In this section, we provide a variation of Algorithm 2 with no fairness constraints but with a budget allocation strategy. See Algorithm 3.

---

**Algorithm 3** Stage 2 Variant: $\mu$-zCDP DP OLS Regression

---

1: **Input**: $Z; \mu; \eta_t, \Delta, T$
2: Obtain standard errors $s_1, s_2, \ldots, s_K$ from Algorithm 1
3: Initialize $\theta_1$ arbitrarily
4: **for** $t = 1$ **to** $t = T$ **do**
5:      **for** $k = 1$ **to** $k = K$ **do**
6:          $\nabla \ell_k(\theta_t, Z) = \sum_{i=1}^{n_k} 2(\langle \theta_t, X_{ki} \rangle - Y_{ki}) X_{ki} = \sum_{i=1}^{n_k} \nabla \ell_k(\theta_t, X_{ki}, Y_{ki})$
7:          Use budget of $\mu_T = \mu \cdot s_k^2 / T$ to compute $M^k$ as follows:
8:          $\nabla L_k(\theta_t, Z) = \sum_{i=1}^{n_k} \frac{\nabla \ell_k(\theta_t, X_{ki}, Y_{ki})}{\max(1, \|\nabla \ell_k(\theta_t, X_{ki}, Y_{ki})\|_2 / \Delta)}$   ($\ell_2$-norm per-example gradient clipping)
9:          $M^k = \nabla L_k(\theta_t, Z) + \mathcal{N}(0, I_{P \times P} \cdot \frac{\Delta^2}{2\mu_T})$
10:      **end for**
11:      $M = (M^1, \ldots, M^K) \in \mathbb{R}^{P \times K}$
12:      $\theta_{t+1} = \Pi_{\mathcal{C}_P}(\theta_t - \eta_t(M \mathbf{1}_K))$
13: **end for**
     **return** $\theta_t$ // public DP release

---

**Lemma C.3.** *For any $\mu > 0$, Algorithm 3 is $\mu$-zCDP.*

*Proof.* We apply Lemma 2.3 in (Bun & Steinke, 2016) (for composition and post-processing of zCDP) and Lemmas 2.4, 2.5 for the zCDP guarantees via the application of the Gaussian Mechanism.

Each iteration $t \in [T]$ receives privacy budget of $\mu/T$. From stage 1, $\sum s_k^2 = 1$. We can use this to allocate budget of $s_k^2 \cdot \mu/T$ for each group $k \in [K]$. By post-processing and composition properties of zCDP, the whole procedure satisfies $\mu$-zCDP.

$\square$

## D   Mixture Model Analysis for Standard Error and Other Proof Details

In this section, we include proofs that we could not, because of space constraints, include in the main paper.

**Lemma D.1.** *Let $U$ be an $\ell_2$-norm upper bound on the rows of $X \in \mathbb{R}^{n \times d}$ and the rows of $Y \in \mathbb{R}^n$. Then $\widetilde{X^T X} = X^T X + \frac{U^2}{\sqrt{2\tau}} Z$ is $\tau$-zCDP where $Z \in \mathbb{R}^{d \times d}$ is a symmetric matrix with each entry in the upper triangular matrix sampled from $\mathcal{N}(0,1)$ and the lower triangle entries are copied from its upper triangle part.*

*Furthermore, $\widetilde{X^T Y} = X^T Y + \frac{U^2}{\sqrt{2\tau}} Z$ is $\tau$-zCDP where $Z \sim \mathcal{N}(0, I_{d \times d})$.*

*Proof.* The proof of privacy follows from the Analyze-Gauss algorithm (see Algorithm 1 and Theorem 2 in (Dwork et al., 2014)) where we apply the Gaussian Mechanism to satisfy $\tau$-zCDP, by Proposition 1.6 in (Bun & Steinke, 2016). $\square$

**Lemma D.2** (Restatement of Lemma 3.1)**.** *For any $c \in (0,1)$, $\beta_1, \beta_2 \in \mathbb{R}^d$, $X_1 \in \mathbb{R}^{n_1 \times d}, X_2 \in \mathbb{R}^{n_2 \times d}$, $\sigma_1, \sigma_2 > 0, n_1, n_2 \in \mathbb{Z}_+$, define the mixture model $Y \sim c\mathcal{N}(X_1\beta_1, \sigma_1^2 I_{n_1 \times n_1}) + (1-c)\mathcal{N}(X_2\beta_2, \sigma_2^2 I_{n_2 \times n_2}) \sim cY_1 + (1-c)Y_2$.*

*Let $\beta = \beta_1$ with probability $c$ and $\beta = \beta_2$ with probability $1 - c$, then the prediction $\hat{Y}$ is $Y_1$ with probability $c$ and $Y_2$ with probability $(1-c)$. The expected prediction loss of $\beta$ on $Y$ is $c^2 \cdot n_1 C \sigma_1^2 + (1-c)^2 \cdot n_2 C \sigma_2^2 + c(1-c)\Gamma_1 + c(1-c)\Gamma_2$ where $\Gamma_1 = \|Y_2 - X_2\beta_1\|^2, \Gamma_2 = \|Y_1 - X_1\beta_2\|^2$, $C \sim \chi_1^2$, and $\chi_k^2$ is the chi-squared distribution with $k$ degrees of freedom.*

*Proof.* First note that $\|Y_1 - X_1\beta_1\|^2 \sim n_1\chi_1^2\sigma_1^2$ since $Y_1 - X_1\beta_1 \sim \mathcal{N}(0, \sigma_1^2 I_{n_1 \times n_1})$ so that for any $i \in [n_1]$, $(Y_1 - X_1\beta_1)_i \sim \mathcal{N}(0, \sigma_1^2) \sim \sigma_1\mathcal{N}(0, 1)$. $(Y_1 - X_1\beta_1)_i^2 \sim \sigma_1^2\chi_1^2$ since $\chi_k^2$ is the sum of squares of i.i.d. standard normal random variables. Then $\|Y_1 - X_1\beta_1\|^2 \sim n_1\sigma_1^2\chi_1^2$. Similarly, $\|Y_2 - X_2\beta_2\|^2 \sim n_2\sigma_2^2\chi_1^2$.

Then the expected loss given the learned estimate $\beta$ is

$$\mathbb{E}_\beta[\|\hat{Y} - Y\|^2] = \mathbb{E}_\beta[c\|Y_1 - X_1\beta\|^2 + (1-c)\|Y_2 - X_2\beta\|^2]$$
$$= c^2\|Y_1 - X_1\beta_1\|^2 + (1-c)^2\|Y_2 - X_2\beta_2\|^2 + c(1-c)\|Y_1 - X_1\beta_2\|^2 + c(1-c) \cdot \|Y_2 - X_2\beta_1\|^2$$
$$\sim c^2 \cdot n_1 C\sigma_1^2 + (1-c)^2 \cdot n_2 C\sigma_2^2 + c(1-c)\Gamma_1 + c(1-c)\Gamma_2,$$

where $C \sim \chi_1^2$. $\square$

As shown above, even the best randomized predictor has dependence on group size and the standard error of the prediction. Compounding this issue, the requirement for differential privacy necessitates additional noise, which increases measurement error. To account for measurement error, we need to make sure that the error in the estimates of $\Gamma_1, \Gamma_2$ are not exacerbated.

# E   Asymptotic Convergence Analysis of (Stochastic) Gradient Descent for CGOO Problem

In this section, we give convergence guarantees for (stochastic) gradient descent for the CGOO problem. We discuss the optimal setting of the learning rate when applying differentially private gradient descent to solve constrained maximization problems defined on smooth functions for the CGOO problem. Assume that the functions $f$ and $g$ have smoothness parameters $\beta_f$ and $\beta_g$ respectively. Recall that a function $f : \mathcal{V} \to \mathbb{R}$ is $\beta$-smooth if its gradient $f$ is $\beta$-Lipschitz. That is, $\|\nabla f(x) - \nabla f(x')\| \leq \beta\|x - x'\|$ for all $x, x' \in \mathcal{V}$.

In Algorithm 4, we reproduce Algorithm 2 but with a learning rate that depends on the privacy noise parameter $\sigma$. We show that, after at most $T = O(n)$ calls to a perturbed version of $\nabla h$, we can differentially privately minimize $h$ to within $\alpha$ for any $\alpha > 0$ for large enough sample size $n$. Note that we could still minimize $h$ to within $\alpha$ with smaller $T$ in practice.

---

**Algorithm 4** $\rho$-zCDP DP (Fair) Regression on Smooth Functions

**Input**: $D, n, \rho, T, \beta_f, \beta_g, G$
$\sigma^2 = \frac{K(\beta_f + G\beta_g)^2 T}{2\rho n^2}$
Arbitrarily set $c_1 \in \mathcal{C}_P$
**for** $t = 1$ **to** $t = T - 1$ **do**
    $\nabla h(c_t, D) = \nabla\ell(c_t, D)\nabla f(\ell(c_t, D)) + \mathbb{1}[g(\ell(c_t, D)) \geq 0]\nabla\ell(c_t, D)\nabla g(\ell(c_t, D))$
    $q_t \sim \mathcal{N}(\nabla h(c_t, D), \sigma^2 I_{P \times P})$
    $\eta(t) = \frac{\sqrt{P}}{\sqrt{t}\sqrt{(1+G)^2 + P\sigma^2}}$
    $c_{t+1} = \Pi_{\mathcal{C}_P}(c_t - \eta(t)q_t)$
**end for**
**return** $c_T$

---

In Figures 7-A, we illustrate (graphically) how projected gradient descent behaves under differential privacy constraints. The projection step in gradient descent becomes noisy because of the noisy gradient computation process. For Frank-Wolfe under differential privacy, the vertices of the set $\mathcal{C}$, if convex, appear to be moved or perturbed. We illustrate differentially private Frank-Wolfe in Figure 7-B. Frank-Wolfe is, essentially, an iterative first-order optimization algorithm that replaces the projection step of gradient descent with solving linear approximation sub-problems.

**Lemma E.1** (Composition of $\rho$-zCDP (Bun & Steinke, 2016))). *Let $\mathcal{M} : \mathcal{X}^n \to \mathcal{Y}$ and $\mathcal{M}' : \mathcal{X}^n \to \mathcal{Z}$ be (randomized) algorithms that satisfy $\rho$-zCDP and $\rho'$-zCDP, respectively. Define $\mathcal{M}'' : \mathcal{X}^n \to \mathcal{Y} \times \mathcal{Z}$ where $\mathcal{M}''(x) = (\mathcal{M}(x), \mathcal{M}'(x))$. Then $\mathcal{M}''$ satisfies $(\rho + \rho')$-zCDP.*

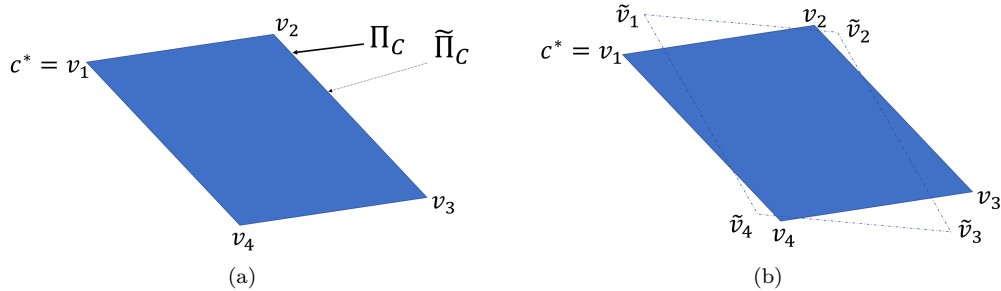

Figure 7: (a) Illustration of Projected Gradient Descent for private Constrained Group-Objective Optimization using a noisy projection oracle. (b) Illustration of perturbed vertices when solving linear optimization subproblems for private Constrained Group-Objective Optimization.

**Lemma E.2** (Post-Processing of $\rho$-zCDP (Bun & Steinke, 2016))). *Let $\mathcal{M} : \mathcal{X}^n \to \mathcal{Y}$ and $g : \mathcal{Y} \to \mathcal{Z}$ be (randomized) algorithms. Suppose that $\mathcal{M}$ satisfies $\rho$-zCDP. Define $\mathcal{M}' : \mathcal{X}^n \to \mathcal{Z}$ as $\mathcal{M}'(x) = g(\mathcal{M}(x))$. Then $\mathcal{M}'$ satisfies $\rho$-zCDP.*

**Theorem E.3** (Theorem 2 from (Shamir & Zhang, 2013)). *Suppose that $h$ is a convex function and that for some constants $S, Q$, it holds that $\mathbb{E}[\|\hat{q}_t\|^2] \leq Q^2$ for all $t$ and $\sup_{c,c' \in \mathcal{C}} \|c - c'\| \leq S$. Then the stochastic gradient descent algorithm with initialization rule $c_1 \in \mathcal{C}$, and update rule*

$$c_{t+1} = \Pi_{\mathcal{C}} \left( c_t - \eta_t \hat{q}_t \right),$$

*and step size $\eta_t = c/\sqrt{t}$ (where $c > 0$ is a constant) has the following guarantee for any $T > 1$:*

$$\mathbb{E}[h(c_T) - h(c^*)] \leq \left( \frac{S^2}{c} + cQ^2 \right) \frac{2 + \log T}{\sqrt{T}},$$

*where $c^* \in \arg\min_{c \in \mathcal{C}} h(c)$, $\hat{q}_t \in \partial h(c_t)$ is a subgradient of $h$ and $\Pi_{\mathcal{C}}$ is a projection oracle into feasible set $\mathcal{C}$.*

**Theorem E.4** (Theorem 3.7 from (Bubeck, 2015)). *On the set $\mathcal{C}$ let projected gradient descent be of the form $c_{t+1} = \Pi_{\mathcal{C}} \left( c_t - \eta \nabla h(c_t) \right)$ for $t > 1$ and $c_1$ be an arbitrary decision in $\mathcal{C}$. Suppose $h$ is convex and $\beta$-smooth on $\mathcal{C}$. Then projected gradient descent with step size $\eta = \frac{1}{\beta}$ satisfies*

$$h(c_t) - h(c^*) \leq \frac{3\beta\|c_1 - c^*\|^2 + h(c_1) - h(c^*)}{t},$$

*where $c^* \in \arg\min_{c \in \mathcal{C}} h(c)$.*

---

**Algorithm 5** Gradient Descent for Constrained Group-Objective Optimization.

---

**Input**: $D, \rho, T, \Pi_{\mathcal{C}}, G$
Arbitrarily set $c_1 \in \mathcal{C}$
**for** $t = 1$ **to** $t = T - 1$ **do**
   $\eta(t) = \frac{\sqrt{K}}{(1+G)\sqrt{t}}$
   $\nabla h(c_t, D) = \nabla \ell(c_t, D) \nabla f(\ell(c_t, D)) + \mathbb{1}[g(\ell(c_t, D)) \geq 0] \nabla \ell(c_t, D) \nabla g(\ell(c_t, D))$
   $c_{t+1} = \Pi_{\mathcal{C}} \left( c_t - \eta(t) \nabla h(c_t, D) \right)$
**end for**
**return** $\hat{c} = c_T$

---

Algorithms 5 and 6 correspond to the algorithms for solving the CGOO problem with learning rates from Theorem E.3 and Theorem E.4 (for the smooth case).

**Lemma E.5.** *For any Lipschitz continuous functions $f, g : [0,1]^K \to \mathbb{R}$, suppose that there exists $\boldsymbol{y} \in [0,1]^K$ such that $g(\boldsymbol{y}) \leq 0$.*

---

**Algorithm 6** Gradient Descent for Constrained Group-Objective Convex Smooth Optimization.

---

**Input**: $D, \rho, T, \Pi_{\mathcal{C}}, G$
Arbitrarily set $c_1 \in \mathcal{C}$
**for** $t = 1$ **to** $t = T - 1$ **do**
  $\eta(t) = \frac{1}{(\beta_f + G\beta_g)}$
  $\nabla h(c_t, D) = \nabla \ell(c_t, D) \nabla f(\ell(c_t, D)) + \mathbb{1}[g(\ell(c_t, D)) \geq 0] \nabla \ell(c_t, D) \nabla g(\ell(c_t, D))$
  $c_{t+1} = \Pi_{\mathcal{C}} (c_t - \eta(t) \nabla h(c_t, D))$
**end for**
**return** $\hat{c} = c_T$

---

*For any $G > 0$, define the function $h : [0,1]^K \to \mathbb{R}$ as follows: $h(\boldsymbol{x}) = f(\boldsymbol{x}) + G \cdot \max(0, g(\boldsymbol{x}))$ for any $\boldsymbol{x} \in [0,1]^K$. Then for all $\boldsymbol{x}' \in [0,1]^K, \alpha > 0$ such that $h(\boldsymbol{x}') \leq \min_{\boldsymbol{x} \in [0,1]^K} h(\boldsymbol{x}) + \alpha$, we are guaranteed that*

$$f(\boldsymbol{x}') \leq \min_{\boldsymbol{x} \in [0,1]^K : g(\boldsymbol{x}) \leq 0} f(\boldsymbol{x}) + \alpha, \qquad g(\boldsymbol{x}') \leq \frac{\alpha + L_f \sqrt{K}}{G}$$

*where $L_f$ is the Lipschitz constant for the function $f$.*

Using Lemma E.5, we can solve the CGOO problem by composition of functions $f, g$ into $h$. So for the rest of this section, we will focus on optimizing the Lagrangian formulation via the function $h$. We assume that $f, g$ are 1-smooth but all theorems can be generalized to handle any $f, g$ with smoothness parameters $\beta_f, \beta_g > 0$.

**Theorem E.6.** *Suppose we are given convex, 1-Lipschitz, 1-smooth functions $f, g : [0,1]^K \to \mathbb{R}$, loss function $\ell : \mathcal{C} \times (\mathcal{X} \times \mathcal{Y} \times \mathcal{A})^n \to [0,1]^K$, and privacy parameter $\rho \geq 0$.*

*Then assuming that*

$$n = \max \left\{ \tilde{\Omega} \left( \frac{KP}{\alpha^4} \right), \Omega \left( \frac{KP}{\rho} \right) \right\},$$

*after $T = O(n)$ iterations, Algorithm 4 is a $\rho$-zCDP differentially private algorithm that returns a decision $\tilde{c} \in \mathcal{C}$ with the following guarantee:*

$$f(\ell(\tilde{c}, D))] \leq f(\ell(c^*, S)) + \alpha, \qquad g(\ell(\tilde{c}, D))] \leq \alpha$$

*with probability at least 9/10 where $c^* \in \arg\min_{c \in \mathcal{C} : g(\ell(c, S)) \leq 0} f(\ell(c, S))$ is the best decision in the feasible set $\mathcal{C} \subseteq \mathbb{R}^P$, $\alpha > 0$. We assume access to a projection oracle $\Pi_{\mathcal{C}}$ and that $\mathcal{C} \subseteq \mathbb{R}^P$ lies in a $P$-dimensional space that is bounded in each axis.*

*Proof.* We combined $f$ and $g$ into a "new" convex function $h(\ell(c, D)) = f(\ell(c, D)) + G\max(0, g(\ell(c, D)))$ defined for all $c \in \mathcal{C}$. Its gradient is given as $\nabla h(c, D) = \nabla \ell(c, D) \nabla f(\ell(c, D)) + \mathbb{1}[g(\ell(c, D)) \geq 0] \nabla \ell(c, D) \nabla g(\ell(c, D))$. Then note that $h$ has smoothness parameter at most $\beta_f + G\beta_g$ if $\beta_f, \beta_g$ are the smoothness parameters for functions $f, g$ respectively.

First, we go through the proof of privacy of Algorithm 4. We invoke Lemma 2.5 in (Bun & Steinke, 2016), which states that if $\mathcal{M} : \mathcal{X}^n \to \mathbb{R}^d$ is a mechanism that releases $\mathcal{N}(q(x), \sigma^2 I_{d \times d})$ for some function $q : \mathcal{X}^n \to \mathbb{R}^d$, then $\mathcal{M}$ satisfies $\rho$-zCDP for

$$\rho = \frac{1}{2\sigma^2} \sup_{D, D'} \|q(D) - q(D')\|_2^2,$$

where $D, D'$ are neighboring databases that differ in one entry.

Then since $\ell(c, D) = \frac{1}{n} \sum_{i=1}^n \ell(c, D_i)$ and $\ell(c, D) \in [0,1]^K$,

$$\|\nabla h(\ell(c, D)) - \nabla h(\ell(c, D'))\| \leq (\beta_f + G\beta_g) \|\ell(c, D) - \ell(c, D')\| \tag{10}$$

$$\leq (\beta_f + G\beta_g) \sqrt{K}/n. \tag{11}$$

As a result, $q_t \sim \mathcal{N}(\nabla h(c_t, D), \sigma^2 I_{P \times P})$ satisfies $\rho$-zCDP (where we split the budget amongst the $T$ rounds).

For the utility guarantees, we will rely on Theorem E.3 which requires a bound $G^2$ on the norm of subgradients $\hat{q}_t$ and a bound on the norm of maximum difference between any two $c, c' \in \mathcal{C}$ which is $\sup_{c,c'} \|c - c'\| \leq \|\mathcal{C}\| = O(\sqrt{P}) = S$. We have that the subgradients are $q_t \sim \mathcal{N}(\nabla h(c_t, D), \sigma^2 I_{P \times P})$ so that

$$\|\nabla h(c_t, D)\|^2 = \|\nabla f(\ell(c_t, D)) + G \cdot \mathbb{1}[g(\ell(c_t, D)) \geq 0] \cdot \nabla g(\ell(c_t, D))\|^2 \tag{12}$$

$$\leq \|\nabla f(\ell(c_t, D))\|^2 + G^2 \|\nabla g(\ell(c_t, D))\|^2 + 2G \langle \nabla f(\ell(c_t, D)), \nabla g(\ell(c_t, D)) \rangle \tag{13}$$

$$\leq \|\nabla f(\ell(c_t, D))\|^2 + G^2 \|\nabla g(\ell(c_t, D))\|^2 + 2G \|\nabla f(\ell(c_t, D))\| \|\nabla g(\ell(c_t, D))\| \tag{14}$$

$$\leq (1 + G)^2 \tag{15}$$

by the Cauchy–Schwarz inequality and the bound on the gradients of $f, g$.

Then since $q_t \sim \mathcal{N}(\nabla h(c_t, D), \sigma^2 I_{P \times P}) = \nabla h(c_t, D) + a_t$ with $a_t \sim \mathcal{N}(0, \sigma^2 I_{P \times P})$ we have that

$$\mathbb{E}[\|\hat{q}_t\|^2] = \mathbb{E}[\|\nabla h(c_t, D)\|^2] + \mathbb{E}[\|a_t\|^2] + 2\mathbb{E}[\langle \nabla h(c_t, D), a_t \rangle] \tag{16}$$

$$\leq (1 + G)^2 + \sigma^2 P, \tag{17}$$

by Fact E.7 so that $\mathbb{E}[\|\hat{q}_t\|^2] \leq Q^2 = (1 + G)^2 + \sigma^2 P$.

Algorithm 4 uses the optimal settings — up to constants — of the learning rate (can be derived via differentiation and solving for the constant $c$) based on Theorem E.3:

$$\eta(t) = \frac{c}{\sqrt{t}} = \frac{S}{Q\sqrt{t}} = \frac{\sqrt{P}}{\sqrt{t}\sqrt{(1 + G)^2 + P\sigma^2}}.$$

Then it follows from Theorem E.3 that the "new" function $h : [0, 1]^K \to \mathbb{R}$ for the setting of $\eta(t)$ has the following convergence guarantees:

$$\mathbb{E}[h(\tilde{c}, D)] - h(\ell(c^*, S)) \leq \frac{2 \log T (2SQ)}{\sqrt{T}} \tag{18}$$

$$= O\left(\frac{\log T}{\sqrt{T}} \left(\sqrt{P}\sqrt{(1 + G)^2 + P\sigma^2}\right)\right) \tag{19}$$

and plugging in $\sigma$ we get

$$O\left(\frac{\log T}{\sqrt{T}} \sqrt{P} \sqrt{(1 + G)^2 + \frac{TPK(1 + G)^2}{\rho n^2}}\right).$$

Setting $T = O(n)$ and using that $n = \Omega(PK/\rho)$, we can further simplify to obtain that

$$\mathbb{E}[h(\tilde{c}, D)] - h(\ell(c^*, S)) \leq \alpha,$$

if $n = \tilde{\Omega}(PK/\alpha^4)$ for $G = \sqrt{K}/\alpha$. And by Lemma E.5 this implies that

$$\mathbb{E}[f(\ell(\tilde{c}, D))] \leq f(\ell(c^*, S)) + \alpha, \qquad \mathbb{E}[g(\ell(\tilde{c}, D))] \leq \alpha.$$

Applying Markov's inequality then gives the desired result.

$\square$

**Fact E.7.** *Let $Z \sim \mathcal{N}(0, \mathbb{I}_{P \times P})$. Then for any (fixed) vector $v \in \mathbb{R}^P$,*

$$\langle Z, v \rangle \sim \mathcal{N}(0, \|v\|_2^2).$$

