# OpenReview forum: "Privacy Budget Tailoring in Private Data Analysis"
_TMLR — Accepted by TMLR_

### Review · Reviewer_BzM5 · 2023-09-15

**Summary Of Contributions:**

This paper proposes a group-aware DP fair regression algorithm to avoid the disparate performance for minority groups in differentially private linear and logistic regression models. Theoretically, this paper uses the notion of zCDP to measure the privacy for the proposed algorithm. Experimentally, a significant reduction in mean square error (MSE) is observed on both synthetic and real-world datasets.

**Audience:**

Yes

**Broader Impact Concerns:**

No concerns about the Broader Impact.

**Claims And Evidence:**

Yes

**Requested Changes:**

------------------ For weakness 1 ------------------
1. The meaning of $\mathcal{A}$ is not clear. On top of page 3: $\mathcal{A}$  the domain of sensitive attributes. However, in Definition 2.2, $\mathcal{A}$ seems to represent the set of all groups. What means "sensitive attributes" and how is it related to the group?

2. The last sentence on Page 6 is lack of reasoning. I guess the authors is trying to use some composition property of zCDP. However, this should not be common knowledge for the readers. A citation and a detailed explanation need to be added.

3. In Algorithms 1 and 2, what means "use budget of ... to sample"? A rigid definition or an explanation should be added.

4. Lots of grammar issues. Especially steps 8 & 9 of Algorithm 1, I think the author intends to say "Use budget of $\tau_k/2$ to sample \tilde{$X^T X$}, which is a differentially private estimation of $X^TX$ (the variance matrix)". Again, I don't know what means "Use budget of $\tau_k/2$ to sample"

------------------ For weakness 2 ------------------

The following two properties in the curve of DPNoSE's MSE in Figure 1 look very suspicious to me
1. It keeps increasing with $\rho$.
2. It seems coverage to a large constant when $\rho\to\infty$

For 1, since a larger $\rho$ corresponds to a milder privacy requirement, usually the MSE is expected to reduce when $\rho$ is larger.

For 2, when $\rho\to\infty$, we no longer need any noise for privacy. If I understand correctly, all four curves should converge to the same (small) constant when $\rho\to\infty$.

------------------ For weakness 3 ------------------

If the complexity of the proposed algorithm is unknown, the mechanism designers will find it hard to decide because most application scenarios have complexity constraints. Either a comprehensive comparison with the standard mechanisms or a rigid theoretical analysis should be able to fill this gap.

------------------ For weakness 4 ------------------
1. On page 7, what means OLS? Is it ordinary least-squares? This abbreviation is never defined in the paper.
2. I don't think the context is clear enough to explain how OLS is applied in the experiments. Thus, further explanation is required. OLS's slopes $\beta_1$ and $\beta_2$ may also require rigid definitions.

------------------ Others ------------------

It looks strange that there are only Lemmas but no Theorem in this paper. I would suggest having a theorem saying that "The entire scheme (Algorithm 1 and 2) is $\rho$-zCDP, where $\rho = \tau+\mu$"



------------------ After rebuttal and revision------------------

Most of my concerns have been addressed. Changed "Claims And Evidence" from "No" to "Yes".

**Strengths And Weaknesses:**

Strength:
  1. The problem studied in this paper is important to the community (Fairness and privacy are both "big concerns" in machine learning)
  2. The experiments observe a significant improvement in comparison with the baseline

Weakness:
  1. The presentation needs huge improvement (hard to follow)
  2. The experimental results in Figure 1 do not make sense to me (additional explanation is required from the authors, see the "requested changes" section). Owing to this, I temporarily put a "No" in "Claims And Evidence". I will change it if the revision and/or rebuttal make sense here.
  3. The complexity analysis of the proposed algorithm is missing.
  4. Some key points are missing in section 5 (maybe this causes weakness 2)

---

> ### Author Response · Authors · 2023-09-17
> **Response to Reviewer BzM5**
>
> Thanks to reviewer BzM5 for the constructive feedback and for identifying the strengths of our work. We have updated our paper to reflect responses to your queries:
>
> **Weakness 1**:
> We have further clarified the meaning of $\mathcal{A}$, the domain of sensitive attributes. As an example that we now included, it could be the possible outcomes of a HIV test. According to the CDC (Center for Disease Control and Prevention), due to state and federal privacy law, the results of a HIV test cannot be released without permission. So one could consider that attribute as sensitive. But zip code might be considered a non-sensitive attribute.
> For (2), indeed, the result relies on the basic composition of differential privacy. We have added an appropriate reference.
> For (3) on using a privacy budget to sample a matrix: By this, we mean that we compute the matrix in a differentially private manner. Since $X^TX$ lives in $\mathbb{R}^{d\times d}$, we are essentially sampling a perturbed version of the original matrix from $\mathbb{R}^{d\times d}$. The sampling procedure can be done using a number of mechanisms (e.g., exponential mechanism or gaussian mechanism) as long as it satisfies differential privacy. We have added additional clarification.
> For (4), we have added further clarification for this and added the proposed sentence.
>
> **Weakness 2**:
> Indeed, as $\rho\rightarrow\infty$, the noise should become negligible. However, all four curves will not necessarily converge to the same constant since our formulation builds on a Lagrangian fair regression which penalizes the larger group so that the learned parameters can result in a smaller MSPE for the minority group.
> For the sake of clarity, we have now replaced figure 1 in the main paper with Tables 1,2,3 that show the behavior of the DPSE and DPNoSE algorithms on 5 groups as $\rho$ increases. For $\rho=2$, the larger group has an MSPE of 1.02 which increases to a larger error of 1.27. But for all the other groups (groups 2,3,4,5), their MSPE decreases. This signifies a “fairer” outcome for the minority groups. As $\rho$ increases, the MSPE decreases for all the groups tends to decrease (as expected). We hope the tables added clarify the outcome of the algorithms.
>
> **Weakness 3**:
> Complexity Analysis: Given that there is always a cost to the use of differential privacy (e.g., see packing-style and fingerprinting lower bounds), we have now added a sample complexity lower bound analysis. These results show the minimum number of samples needed to achieve an accuracy bound while remaining differentially private. In addition, we clarified the time complexity of Algorithms 1 and 2, in terms of the number of groups and the size of each group.
>
> **Weakness 4**:
> We have now explicitly defined OLS and clarified the per-group generation process in terms of the parameters.
>
> **Others**:
> We now have theorems (for the lower bounds) and have clarified the use of basic composition to ensure that the entire scheme is differentially private.

---

### Review · Reviewer_VjZX · 2023-09-19

**Summary Of Contributions:**

The paper proposes a method for improving prediction accuracy for minority groups under DP guarantees. The basic approach is to divide the total available privacy budget unevenly between groups to improve the relative performance of the predictions on minority groups, while (possibly) hurting the performance on the majority. The authors motivate the approach by analysing a simple 2-Gaussian mixture model, and show empirically that the proposed method results in improved prediction performance for the minority group(s). Additionally, the paper includes some additional empirical results with multiple minority groups, as well as some basic convergence theory under some assumptions.

**Audience:**

Yes

**Broader Impact Concerns:**

It is potentially quite problematic to provide weakened privacy guarantees for minority groups.

**Claims And Evidence:**

Yes

**Requested Changes:**

In decreasing order of importance:

1) Glancing at DP release of XX, XY in Lemma C.1, do you assume that 0<= X,Y <=U always? If not, I would think that you need a lower bound on the values as well for the sensitivity (in which case the sensitivity for the cross-terms does not look correct for bounded DP). If yes, please add it explicitly as an assumption.

2) Include some test showing at least a couple of different privacy splits including the one resulting from looking at the std errors to motivate that this is really agood idea (would expect the proposed method to be consistently at least close to the best choice).

3) Maybe reformulate the discussion on the weakening of the minority DP guarantees a bit: might be better to set the target DP guarantees according to the minority protection, while the majority would then get that much more privacy.

4) In summarising the results (e.g., Intro), it would be good to also state the evidence of how much the majority tends to lose. The current wording seems to indicate that this is not a trade-off but simply a gain.



Minor corrections, typos etc. (no need to comment on these)

* There are some bad sentences in the Appendices (e.g. first paragraph in A.2), probably resulting from moving the text from the main body into the Appendix without fixing it in any way.

* The notation for clipping seems bad, since it leaves one guessing which terms are actually included in the clipping operation.

* Sec 3.2, Setup and Notation "formulation for fair requires".

* Algorithm 1, line 8: variance matrix?

* Are the labels mixed in Fig 6? The results look a bit surprising.

* Appendix D: gradient f missing nabla.

**Strengths And Weaknesses:**

### Strengths

* The writing is mostly clear and easy to follow.

* The problem of improving minory group results (under DP) is an important and timely one.

* The proposed approach is simple and seems to work (although see also weaknesses).

### Weaknesses

1) What the authors propose is basically replacing the worse-than-majority prediction performance for the minorities with worse-than-majority privacy protection for the same. I do not agree with the authors that this is acceptable since even very large epsilons have been experimentally shown to protect against some attacks: I would argue that due to the privacy-utility trade-off, DP guarantees will pretty much always be set to be as weak as can be tolerated in order to get the most useful models. In such cases, further weakening the guarantees is not feasible.

2) I am not sure whether it is a good practice to report the DP guarantees as a single guarantee in this case, since we know that the guarantees are different for the majority and the minority groups. To me, this looks like it could very easily give the false impression of having equal privacy protection between the groups.

3) The empirical results, especially for the more complex models, would be more convincing if they included comparisons using various amounts of privacy apportioned between the majority and the minory group, and then showing that the specific amount resulting from looking at the std errors as proposed is a good one compared to all the other possibilities.

4) The analysis only looks at a simple 2-mixture Gaussian, while anything more complex is only tested empirically.

5) The proposed method introduces new hyperparameters that need to be tuned, without much guidance as to how one should set them. This is especially problematic when the tuning needs to be done under DP; e.g., the paper only considers how to divide the privacy budget in the second stage of the proposed algorithm, while the division between stages and division between loss and gradient (in Algorithm 2) are left as free extra hyperparameters.

---

> ### Author Response · Authors · 2023-09-19
> **Response to Reviewer VjZX**
>
> Thanks to reviewer VjZX for the helpful feedback and for articulating where our work excels and how we can improve it. We especially like the suggestion of explaining that one can set a minimum level of protection for the minority subpopulations. We have updated our paper to reflect responses to your queries:
>
> -- *Minor corrections, typos etc. (no need to comment on these)*
>
> We have now gone through all minor corrections and typos. Thank you!
>
> -- *I am not sure whether it is a good practice to report the DP guarantees as a single guarantee in this case, since we know that the guarantees are different for the majority and the minority groups. To me, this looks like it could very easily give the false impression of having equal privacy protection between the groups.*
>
> We have tried to be explicit with the fact that we allocate different privacy budgets to different subgroups so that it is **not** the case that the subgroups have equal privacy protection. However, in the introduction, we have now added more discussion, especially how it relates to the similar methodology of Abowd & Shmutte (2018). In particular, Abowd & Shmutte (2018) treat privacy budget allocation as a resource allocation problem via an economic lens: choose budgets such that the marginal cost of increasing privacy equals the marginal benefit. In our case, we just use statistical properties of the dataset itself (e.g., standard errors) for resource allocation.
>
> -- *Glancing at DP release of XX, XY in Lemma C.1, do you assume that 0<= X,Y <=U always? If not, I would think that you need a lower bound on the values as well for the sensitivity (in which case the sensitivity for the cross-terms does not look correct for bounded DP). If yes, please add it explicitly as an assumption.*
>
> Indeed, we assume that the $\ell_2$ norm of the rows are bounded by $U$. We have added more precision to the statement of the theorem.
>
> -- *Include some test showing at least a couple of different privacy splits including the one resulting from looking at the std errors to motivate that this is really a good idea (would expect the proposed method to be consistently at least close to the best choice).*
>
> In the appendix (Section A.1), we have now included some results and accompanying discussion that show different privacy budget splits as a result of the standard errors for the subgroups. For these experiments, the dataset sizes for the majority and minority groups are quite imbalanced to illustrate a more aggressive budget allocation strategy.
>
> -- *Maybe reformulate the discussion on the weakening of the minority DP guarantees a bit: might be better to set the target DP guarantees according to the minority protection, while the majority would then get that much more privacy.*
>
> We have now added more discussion about the suggested setting (which one of the authors of this work strongly agrees with): set target DP guarantees for the minority sub-groups.
>
> -- *In summarising the results (e.g., Intro), it would be good to also state the evidence of how much the majority tends to lose. The current wording seems to indicate that this is not a trade-off but simply a gain.*
>
> In the introduction, we have added more explanation indicating that this is indeed a tradeoff (see paragraph that starts with “A potential objection…”) We explain that the majority group might lose in predictive accuracy.

---

> > ### Comment · Reviewer_VjZX · 2023-10-18
> > **Clarification about the privacy budget split**
> >
> > Thanks for the rebuttal. I have some further questions about the privacy budget splits:
> > I'm not sure if you understood my question, so what I originally meant was that you can trivially split the privacy budget between the majority and minority groups, e.g., either by some rule-of-thumb or by relative data set sizes, which you assume are public information (and you also know that std error scales with the sample size). What I don't understand from the provided answer and from appendix A.1 is whether having the entire 1. stage to privately estimate the std errors in order to do a privacy budget split is of any use or not.

---

> > > ### Author Response · Authors · 2023-10-18
> > > **About the privacy budget split**
> > >
> > > Thanks for the clarification. Indeed, you can trivially split the privacy budget between the majority and minority groups via some rule-of-thumb that does not use up the privacy budget. We tried this -- obviously, since no budget is used up in the first stage, this rule-of-thumb/non-private splits will be at least as good as the one that uses up the privacy budget (since it allows for the entire privacy budget to be used in the latter stage).
> > > However, for generality purposes (i.e., extending standard error estimation beyond just linear regression), we decided to estimate stage 1 in a private manner too, by using up some budget. In particular, consider the following scenarios:
> > > 1) *The standard deviation in the minority group is very large compared to the majority group*: although we assume that the dataset size (for both the majority and minority group) is public, we do not assume that the standard deviation is. The standard error estimate captures both the dataset sizes and the variance in each group.
> > > 2) *Beyond Linear Regression*: The standard error indicates the uncertainty of the coefficients. For other models, only the size of the dataset will not capture the uncertainty.
> > >
> > > However, we agree that for simple models where the majority/minority groups have the same or similar uncertainty, using a rule-of-thumb just based on the dataset sizes suffices.

---

> > > > ### Comment · Reviewer_VjZX · 2023-10-18
> > > > **A further comment**
> > > >
> > > > Thanks for the clarification, the second point 2. is mainly why I would like to see some experiment to actually demonstrate that estimating the std errors gives you something beyond just looking at the data set sizes. I still think that providing some such example would make the paper clearly stronger.

---

> > > > > ### Author Response · Authors · 2023-10-18
> > > > > **Additional experiments on why we look beyond data set sizes**
> > > > >
> > > > > -- *I would like to see some experiment to actually demonstrate that estimating the std errors gives you something beyond just looking at the data set sizes.*
> > > > >
> > > > > We have now added additional experiments to illustrate why looking at the standard errors
> > > > > could provide information that is **impossible** to obtain by just looking at the
> > > > > data set sizes. In particular, we provide datasets where two groups have **exactly**
> > > > > the same data set sizes. So just looking at the sizes alone would result in the
> > > > > same privacy split. However, one group has much larger sampling error and the other does not.
> > > > > According to our methodology, more budget would be allocated to the group with larger
> > > > > sampling error. We have now added these experiments to the appendix and a paragraph
> > > > > titled "Why Even Look at Standard Errors?" to that section.
> > > > > We consider the following data set sizes (for both the majority and minority group):
> > > > > 1000, 50000, 9000000.
> > > > >
> > > > > We vary the variance in the independent variable as follows:
> > > > > the first group has variance 10000 and the second has variance of 10 (much smaller!).
> > > > > To account for the severe imbalance in the variance, our methodology assigns a privacy
> > > > > budget split that favors the group with much larger variance.
> > > > > For completeness, we also post the calculated standard errors in the paper.

---

> > > > > > ### Comment · Reviewer_VjZX · 2023-10-20
> > > > > > **Thanks for the update**
> > > > > >
> > > > > > While I would much prefer seeing the performance of the proposed method on running some actual models on data, I think the provided numbers do present a minimal example for the case when estimating the standard errors could provide something usefull beyond the sample size.

---

### Review · Reviewer_cZaT · 2023-10-14

**Summary Of Contributions:**

The paper aims to mitigate the disparate performance for differential private linear regression. Specifically, the authors design a two-stage approach: (1) tailor the privacy budget to the different groups, (2) use linear optimization oracles in a grid to optimize Lagrangian objectives that correspond to fair learning and optimization. Empirical results demonstrate that the proposed group-aware budget allocation method effectively reduces the disparate performance between different groups on synthetic and real-world datasets.

**Audience:**

Yes

**Claims And Evidence:**

Yes

**Requested Changes:**

Please address the Weaknesses and minor problems in the above section.

**Strengths And Weaknesses:**

Strengths
1. The paper is well-written.
2. The problem is well-motivated.
3. The paper is technically sound.

Weaknesses:
1. The organization of the paper is a bit messy. In particular, the experiment results could be more organized, and the results findings should be highlighted. In the Sec. 6 empirical evaluation, the organizations are divided into multiple paragraphs, with some of the main results deferred to the appendix. This makes the section hard to read. In addition, taking Sec. 6.1 “Increasing Number of Groups” as an individual subsection is strange since this organization will diverge the reader’s attention from the main results of synthetic and real-world credit card datasets.
2. Empirical baselines are missing. In Sec. 1, “Introduction,” the authors mentioned a series of closely related works (e.g., Abowd & Schmutte (2018) and Xu et al. (2020). I believe that the authors also need to compare their methods empirically and understand the pros and cons of each method compared to the proposed method.


Minor problems:
1. Theoretical analysis in Sec. 3.1 needs to be improved: In “Why allocate privacy budget based on standard errors?”, I suggest providing a more rigorous way to present this part (e.g., lemmas).

---

> ### Author Response · Authors · 2023-10-15
> **Response to Reviewer cZaT**
>
> Thanks to reviewer cZaT for the positive feedback and for providing specific points we can address and improve upon. We have updated our paper to reflect responses to your queries:
>
> -- *The organization of the paper is a bit messy ... .In addition, taking Sec. 6.1 “Increasing Number of Groups” as an individual subsection is strange since this organization will diverge the reader’s attention from the main results of synthetic and real-world credit card datasets.*
>
> We have transformed Sec. 6.1 for clarity sake (it is no longer its own individual subsection). We have also grouped all experimental results about group sizes and number of groups into its own paragraph. To remain within the 12-page TMLR main body limit for our submission track, we put highlights of our experimental results in Section 6 and refer the reader to the appendix for additional experimental details.
>
> -- *Empirical baselines are missing. In Sec. 1, “Introduction,” the authors mentioned a series of closely related works (e.g., Abowd & Schmutte (2018) and Xu et al. (2020). I believe that the authors also need to compare their methods empirically and understand the pros and cons of each method compared to the proposed method.*
>
> We have included the following empirical baseline comparison:
>
> Bagdasaryan et al. (2019): We replicate the MNIST "8-vs-2" binary classification experiment from the work of Bagdasaryan et al. (2019). We use a logistic regression model in the first stage, and a LeNet network for the second stage (total $\epsilon=5.73$). This means that we tailor the group privacy budgets based on the residuals of the logistic regression for the LeNet model. We find a 64 percent reduction in classification error, for the minority "8" class, compared to the results of Bagdasaryan et al. (2019). This experiment suggests that our approach can generalize to other model classes.
>
> Comparison to Abowd & Schmutte (2018):
> The authors leverage economic analysis to handle budget allocation as a resource allocation problem. Specifically, they operate where the marginal cost of increasing privacy equals the marginal benefit. To calculate the marginal cost of increasing accuracy, in terms of foregone privacy protection, they find the partial derivatives with respect to the privacy parameter and the accuracy. Now using this formula, one can calculate the marginal rate of transformation, which measures the cost of increased loss of privacy and can also measure the social willingness to accept privacy loss, both measured in units of increased statistical accuracy. We note that these quantities cannot be directly estimated from just a statistical model. Thus, even though we are inspired by the work of Abowd and Schmutte, it is not possible to directly empirically compare our model to their work without additional information that measures the social willingness to accept privacy (e.g., via survey collection). We leave such economic/social-scientific work to future work.
> We have also now included further text in our paper to compare and contrast to the work of Abowd & Schmutte (2018).
>
>
> -- *Theoretical analysis in Sec. 3.1 needs to be improved: In “Why allocate privacy budget based on standard errors?”, I suggest providing a more rigorous way to present this part (e.g., lemmas).*
>
> Lemma 3.1 (stated as Lemma C.2 in the appendix and with an accompanying proof) provides an argument (in terms of a Lemma) for why we allocate privacy budget based on standard errors. We have now rewritten parts of the argument to clearly state the relevance of Lemma 3.1 to the privacy allocation strategy. In addition, at the end of Section 3.1, we have added another paragraph with formal remarks about using standard errors for privacy allocation.

---

### Comment · Action_Editors · 2023-11-09
**Some further clarifications**

Dear authors

Before my final decision, I would like to still ask couple of questions and make some remarks regarding the manuscript.

**Regarding Section 3.1**: You motivate the standard deviation based budget allocation through the mixture model example. In the beginning of this Section, you do make couple of remarks that I would like to be clarified.
1. "Learning predictors with low error, in this case, requires randomization (Kalai et al., 2010)": I don't quite understand this remark, and don't see how the attached reference is relevant.
2. "To learn a single predictor when group membership is unknown, the best predictor is one that predicts $\beta_1$ with probability c and $\beta_2$ with probability (1 − c)": Is this always true? Could you add a reference for this claim (or alternatively provide a proof)? **This is an important  clarification**, as this example gives the theoretical justification for your budget splitting.
3. "We wish to minimize the $\ell_2$ loss of a differentially private regression estimate, across these groups, which is equivalent to minimizing": I don't quite understand the equivalence. More specifically, is the DP estimate now obtained through DPGD?
4. "... we have the inequality in expectation:": are the $L_1$ and $L_2$ now the same as defined above? If so, I don't understand how this inequality holds in expectation. We are adding zero-mean Gaussian noise, so shouldn't there be an equality instead of an inequality in expectation?
5. "For large enough clipping parameter, the estimator $F_2(\Delta, \rho)$ will be more accurate if $N_2(\Delta, \rho)$ does not overwhelm the signal in $A$": more accurate than what, the $F_1$? Or are you just saying that $F_1$ is more accurate if the signal is stronger than the noise?

**Regarding Section 4**
1. In Alg. 1, the step 19 returns this data independent vector if the (DP) estimated sum of squared errors is $<0$. I just want to verify that the algorithm indeed terminates here, and it's not just that you set $s_k = 1 / \sqrt{K}$.
2. Continuing on this exception: Do you know how often this happens in practice? I would imagine that for small groups and strong DP guarantees, this exception would be very likely, and in this case (I believe) your Alg. 2 would allocate equal budget for all groups.
3. The privacy guarantee of Alg. 2 is stated bit weirdly in Lemma 4.2. To me, it seems that the Alg. 2 uses the entire $\mu$ budget between lines 5 and 19. Since the line 3 also needs a privacy budget, it should be explicitly stated in Lemma 4.2.
4. A very minor point: It would be helpful for the reader if you could restate at some point, that the $K$ splits of the data can overlap. If they cannot overlap, I don't think there is any reason to split the budget across groups like you do in Alg. 2.

**Regarding Section 5**: I have bit hard time understanding the motivation of $f$ and $g$ in this Section. They seem completely disconnected from the regression case you study. It would be good if you could add more motivation for this Section, and how it relates to the problem you study.

**Regarding Section 6**: You have majority of the results in the Appendix, which makes the presentation bit awkward for the reader. However, more importantly, there are several results that don't have appropriate pointers in the main text. E.g. the paragraph "Beyond Linear Regression." discusses the results, but does not have any pointer where to search the results. Also Table 1 is not referenced in the text at all.
1. Minor point: In Table 2 you say "All results in the table are scaled by 10e−5". Is the MSPE for the DPNoSE really 600 thousand times larger than DPSE for the last row?

---

> ### Author Response · Authors · 2023-11-09
> **Response to Action Editor**
>
> Thanks to the action editor for the remarks. We now respond to your comments/concerns:
>
> **Regarding Section 3.1**:
> Overall, we will clarify and sharpen the writing in Section 3.1.
>
> *"Learning predictors with low error, in this case, requires randomization (Kalai et al., 2010)"*: We just mean that deterministically basing predictions on one component when group membership is unknown can result in large error. In that paper [1], they consider the Gaussian Mixture Model (GMM) with density $F = w_1F_1 + w_2F_2$ where $w_1+w_2=1$ and $F_1, F_2$ are the densities of $\mathcal{N}(\mu_1, \Sigma_1)$ and $\mathcal{N}(\mu_2, \Sigma_2)$ respectively. Note that this is similar to the (isotropic) setting we consider in Section 3.1. But in our case $\mu_1 = X_1\beta_1$ and $\mu_2 = X_2\beta_2$. The paper shows statistical-distance lower bounds between 2 gaussians. As such, when there is a mixture, we cannot deterministically base our predictions on one of the components (since that would increase the statistical distance, resulting in a larger error). For example, Algorithm 1 in [1] chooses a “uniformly random orthonormal basis” when estimating the parameters of the distribution. And as they show in that paper, that strategy results in the optimal approximation (under certain conditions).
>
> *"To learn a single predictor when group membership is unknown, the best predictor is one that..."*:
> This is not always true if the distribution is not like the one specified in the paper. However, for two gaussians (for simplicity consider the univariate case) of the form $\mathcal{N}(\epsilon, 1)$ and $\mathcal{N}(0, 1)$, the statistical distance is at least $\epsilon/20$ (e.g., see [1]). As a result, to close the prediction error, the means should be as close as possible (or the difference between the means should be as small as possible). We can add citations for this result.
>
> *"We wish to minimize the $\ell_2$ loss of a differentially private regression estimate, across these groups, which is equivalent to minimizing: I don't quite understand the equivalence. More specifically, is the DP estimate now obtained through DPGD?*
> Yes, this example is assuming that the DP estimate is obtained via DPGD. We can clarify this point.
>
> *"... we have the inequality in expectation:"*:
> Yes, you’re right: this should say “with high probability” so that the inequality holds (via standard tail bounds of the Gaussian which we can include citations for).
>
> *"...are you just saying that $F_1$ is more accurate if the signal is stronger than the noise?"* Yes.
>
> **Regarding Section 4**
>
> *I just want to verify that the algorithm indeed terminates here, and it's not just that you set $s_k = 1 / \sqrt{K}$.*
>
> This can be a choice of the algorithm designer. The promise of the algorithm is to return DP standard errors. So when it is negative (which shouldn’t be possible without DP noise addition), we can halt/terminate or return “no information” via the setting $s_k = 1 / \sqrt{K}$ (this is equivalent to equal split of the budget between the groups).
>
> *"Continuing on this exception: Do you know how often this happens in practice? I would imagine that for small groups and strong DP guarantees, this exception would be very likely"*
> This is very data dependent. In particular, the DP estimates are more likely to be $\leq 0$ if the standard error is small. So you’re right, for small groups and strong DP guarantees, this would be likely.
>
> *The privacy guarantee of Alg. 2 is stated bit weirdly in Lemma 4.2. To me, it seems that the Alg. 2 uses the entire $\mu$ budget between lines 5 and 19. Since the line 3 also needs a privacy budget, it should be explicitly stated in Lemma 4.2.*
>
> Ok, that sounds good. We can explicitly state this in Lemma 4.2.
>
> *A very minor point: It would be helpful for the reader if you could restate at some point, that the $K$ splits of the data can overlap. If they cannot overlap, I don't think there is any reason to split the budget across groups like you do in Alg. 2.*
>
> Indeed, the groups can overlap (so we cannot rely on parallel composition). If no overlap occurs, then there’s no need to split the budget. We can clarify this point.
>
> **Regarding Section 5**:
> We can add further motivation for this section. You can think of $f$ as the average error across the groups and $g$ as the worst-case error (e.g., max over the groups). We will further clarify this point.
>
> **Regarding Section 6**:
> Unfortunately, we were constrained by the 12-page TMLR limit and placed a large swath of our experimental details in the appendix, as a result. The “Real-World Credit-Card Dataset” is extremely skewed and values normalized. We will remove the use of this dataset and instead focus on the synthetic data experiments and on the law school dataset (which is way less skewed!).
>
> [1] Adam Tauman Kalai, Ankur Moitra, and Gregory Valiant. Efficiently learning mixtures of two gaussians. 2010.

---

> > ### Comment · Action_Editors · 2023-11-10
> > **Continuing on Section 3.1**
> >
> > Thanks for the response!
> >
> > I would like to still ask about the optimality of the proposed predictor. For simplicity, let's focus on the univariate Gaussian case you used in your response. So let's say that we have a Gaussian mixture $Y \sim c N(0, 1) + (1-c) N(\epsilon, 1)$. Now, say that we have a use a predictor $\hat{Y}$ for $Y$ which predicts $0$ w.p. $q$ and $\epsilon$ w.p. $1-q$. Therefore you would get the predictor used in the paper by setting  $q = c$. Now, it is rather easy to see that $Y-\hat{Y}$ is again a mixture of Gaussians: $Y-\hat{Y} \sim qc N(0, 1) + (1-q)c N(-\epsilon, 1) + q(1-c)N(\epsilon, 1) + (1-q)(1-c)N(0, 1)$. Therefore, the MSE of this random variable would be
> > \begin{align}
> > \mathbb{E}[(Y-\hat{Y})^2] &= qc + (1-q)c (1 + \epsilon^2) + q(1-c)(1+\epsilon^2) + (1-q)(1-c) = 1 + c \epsilon^2 + q \epsilon^2 - 2 c q \epsilon^2 = 1 + c \epsilon^2 + q (\epsilon^2 - 2 c  \epsilon^2).
> > \end{align}
> > From the above expression, we see that the MSE is a line w.r.t to $q$, and it is either ascending or descending based on the sign of $1-2c$. This would seem to imply, that the optimal (w.r.t MSE) probability for the predictor would be either $q=0$ or $q=1$.
> >
> > If I'm not mistaken, and the simple analysis above extends to your case, this would seem to suggest that there are other (simple) predictors that reduce the MSE compared to $q=c$ case.
> >
> > Please let me know if I have made an error in my analysis, or if the optimality you consider is somehow fundamentally different.

---

> > > ### Author Response · Authors · 2023-11-10
> > > **Gaussian Mixture Optimality**
> > >
> > > As our paper is not focused on gaussian mixture optimality guarantees, we could remove the claims about optimality (in reference to "Learning predictors with low error, in this case, requires randomization") just so we do not distract the reader from our main results.
> > >
> > > However, we are not sure about how the expression $\mathbb{E}[(Y - \hat{Y})^2]$ was calculated since $qc\mathcal{N}(0, 1)\sim\mathcal{N}(0, (qc)^2))$?
> > >
> > >  (using that $Z\sim\mathcal{N}(0, 1)$ implies that $X = \sigma Z + \mu \sim \mathcal{N}(\mu, \sigma^2)$).

---

> > > > ### Comment · Action_Editors · 2023-11-10
> > > > **About dropping the optimality claim**
> > > >
> > > > I fully agree with the authors, that since the paper is not about building optimal predictors for GMMs, it makes sense to drop the claim and indeed focus on the main results of the paper. I just wanted to point this out, as the Lemma 3.1 seems to rely on this particular predictor, and later the Lemma 3.1 is used as a motivation for the std. based budget allocation (in the paragraph before Section 3.2.).
> > > >
> > > > As a final small remark on my previous example: apologies for the possibly confusing notation. I used $qc N(0,1)$ to denote a component of the Gaussian mixture which has probability $qc=\Pr(\hat{Y}=0)\Pr(Y \text{ is sampled according to the first component})$. The expectation I had above was over both $Y$ and $\hat{Y}$.

---

### Decision · Action_Editor_sN35 · 2023-11-21

**Recommendation:** Accept with minor revision

**Comment:**

Given the fact that the paper presents a novel algorithm to boost fairness in differentially private learning, and that the empirical evidence shows that the algorithm can provide significant improvements, I believe the paper should be accepted. This sentiment was also generally shared among the reviewers, although one of the reviewers didn't vote for acceptance.

However, the paper still needs some polishing before publication. As noted by the reviewers BzM5 and cZaT in their original reviews, the organization of the paper needs some improvements. After reading the updated manuscript, I agree with the reviewers on this. For me the most crucial parts that need to be clarified before acceptance are:
1. Section 3.1: how the Lemma 3.1 connects to the rest of the Section. More precisely, how does the Lemma 3.1 motivate the paragraph "Why allocate privacy budget based on standard errors?".
2. Clarify the importance of Section 5, and how it connects to the rest of the paper.
3. Consider moving the "Helper Lemmas" to the appendix, and moving the empirical evidence from appendix back to the main paper. I think this would highly benefit the reader.
4. Provide the clarifications you have promised post-rebuttal. Specifically the ones listed in response to [my comment](https://openreview.net/forum?id=SnPEhMyuYX&noteId=UvTlrPKIG6).

**Audience:**

Combining differential privacy and fairness is a topic of great importance. While the paper mainly studies simple problems (linear/logistic regression), I still expect this paper to be interesting to the TMLR's audience, and to provide a good starting point for any future work on the domain.

**Claims And Evidence:**

The main claim of this paper, is that the proposed privacy budget allocation scheme alleviates the fairness gap between two sub-groups of the data. Authors test the proposed method in various settings on both real and synthetic data. The support for the claims from the observations of these test is two-fold: the proposed method improves the prediction performance for both the majority/minority groups and the proposed method reduces the accuracy gap between the two groups. While the majority of the empirical evidence is based on tests on synthetic data, I think the evidence is strong enough to support the claims.